# Prioritized Shortest Path Computation Mechanism (PSPCM) for wireless sensor networks

**Innocent Uzougbo Onwuegbuzie**[1,2]*, **Shukor Abd Razak**[1], **Ismail Fauzi Isnin**[1], **Arafat Al-dhaqm**[1], **Nor Badrul Anuar**[3]

**1** School of Computing, Faculty of Engineering, Universiti Teknologi Malaysia, Skudai, Johor, Malaysia, **2** Department of Computer Science, The Federal Polytechnic Ado-Ekiti, Ekiti State, Nigeria, **3** Department of Computer System and Technology, Faculty of Computer Science and Information Technology University of Malaya, Kuala Lumpur, Malaysia

* iuonwuegbuzie@graduate.utm.my

**Data Availability Statement:** All relevant data are within the paper and its Supporting information files.

## Abstract

Routing Protocol for Low-power and Lossy Networks (RPL), the de facto standard routing protocol for the Internet of Things (IoT) administers the smooth transportation of data packets across the Wireless Sensor Network (WSN). However, the mechanism fails to address the heterogeneous nature of data packets traversing the network, as these packets may carry different classes of data with different priority statuses, some real-time (time-sensitive) while others non-real-time (delay-tolerant). The standard Objective Functions (OFs), used by RPL to create routing paths, treat all classes of data as the same, this practice is not only inefficient but results in poor network performance. In this article, the Prioritized Shortest Path Computation Mechanism (PSPCM) is proposed to resolve the data prioritization of heterogeneous data and inefficient power management issues. The mechanism prioritizes heterogeneous data streaming through the network into various priority classes, based on the priority conveyed by the data. The PSPCM mechanism routes the data through the shortest and power-efficient path from the source to the destination node. PSPCM generates routing paths that exactly meet the need of the prioritized data. It outperformed related mechanisms with an average of 91.49% PDR, and average power consumption of 1.37mW which translates to better battery saving and prolonged operational lifetime while accommodating data with varying priorities.

## 1. Introduction

The world is technologically changing at a phenomenal rate, the birth of artificial intelligence and the Internet of Things (IoT) and Fifth Generation (5G) technology [1–3] are reshaping how we live our everyday lives. As it stands, information is now readily available right on our palms, fingertips, or at the push of a button. Intelligent sensors, cooperatively working together forming a network called Wireless Sensor Networks (WSN) [4–6] installed in target locations called sensor field, monitoring physically changing parameters that are both time-sensitive

**Funding:** The author(s) received no specific funding for this work.

**Competing interests:** The authors have declared that no competing interests exist.

and delay-tolerant, that is, real-time and non-real-time activities. The combination of these types of data in the same data stream is referred to in this article as heterogeneous data. Sensed data are routed to locations where they are stored or processed for immediate use. In 2012, the Internet Engineering Task Force (IETF) released the Routing Protocol for Low-power and Lossy network (RPL) [7–9] as the standard routing protocol for resource-constrained devices which form the building blocks for IoT. The routing paths established by RPL are formed by Objective Functions (OF). Currently, RPL has two OFs; Objective Function Zero (OF0) [3] and Minimum Rank with Hysteresis Objective Function (MRHOF) [3]. These OFs are implemented by either link metrics, node metrics, or a combination of both.

While OF0 uses a minimum number of hop-counts (HC) to decide the shortest routing path amongst other paths to the sink, MRHOF uses the Expected Transmission Cost (ETX) to establish the least path cost to the sink. However, establishing these paths, no consideration is given to the heterogeneous nature of data streaming through the network. The deployment of WSN is mostly designed to monitor infrastructures that generate heterogeneous data such as in Wireless Body Area Network (WBAN), smart home, industry, agriculture, logistics, multimedia streaming, smart cities, and in the military where real-time, non-real-time, or the combination of both types of data are generated either in the same or different sensor field. However, the standard routing protocol—RPL, does not consider data heterogeneity while creating routing paths as attention is focused on energy-efficient routing for a prolonged operational lifetime which is due to the power-constrained nature of the sensor nodes [4, 9]. In this article, data heterogeneity represents the nature of the data traversing the network, that is, time-sensitive and delay-tolerant data. To truly accommodate the heterogeneous nature of data streaming through WSN, it becomes imperative to prioritize data packages such that the routing mechanism can handle data according to their priorities without losing sight of power efficiency while giving data packets the timely attention deserved.

Request for Comments (RFC) 6550 describes RPL as a low power consumption distance-vector proactive routing protocol for resource-constrained devices that form the building block for IoT [10]. RPL sets up a self-organized, self-healing, self-sustained, and loop-free routing graph known as *Distance Oriented Directed Acyclic Graph (DODAG)* [11, 12]. DODAG forms a virtual web of connected resource-constrained nodes (that is, constrained by power, storage, and processing capabilities), sensing their immediate environment and cooperatively relaying the sensed data in a hop-by-hop manner to a more resource robust device that acts as the network controller and manager called the *Sink* or *Root node* (*Sink* will be used in this article) [3, 13]. The sink acts in the dual capacity of managing the WSN and relaying sensed data to more robust networks outside the WSN for data storage or further data processing. In comparison to other nodes, the sink is resource enriched in power, storage, and processing capacities. Connecting hundreds and possibly thousands of devices, RPL enabled WSN nodes are expected to last unattended for several months or even years. With the emergence of 5G, which features very low latency, as low as 1milisecond, and very high bandwidth and throughput [14, 15], it becomes imperative to carefully and consciously develop a routing system that is capable of handling data of varying priorities, without compromising the integrity of the network. Fig 1 shows the RPL routing architecture and applications of WSN with 5G [16].

With a data rate of 250kbps [17, 18], RPL wireless medium is termed lossy, due to its high Bit Error Rate (BER) and weak throughput, hence the design and implementation of RPL must be mindful of this weakness as best effort is used to sustain the propagation of data across the lossy network by constantly providing alternate routing paths needed to establish reliable and stable routes from the source node to the sink. For a consistent and sustained operational lifetime, the tree-like network topology of DODAG is set up in such a way that each node selects a power-efficient path and parent node to transport its data in a hop-by-hop manner to

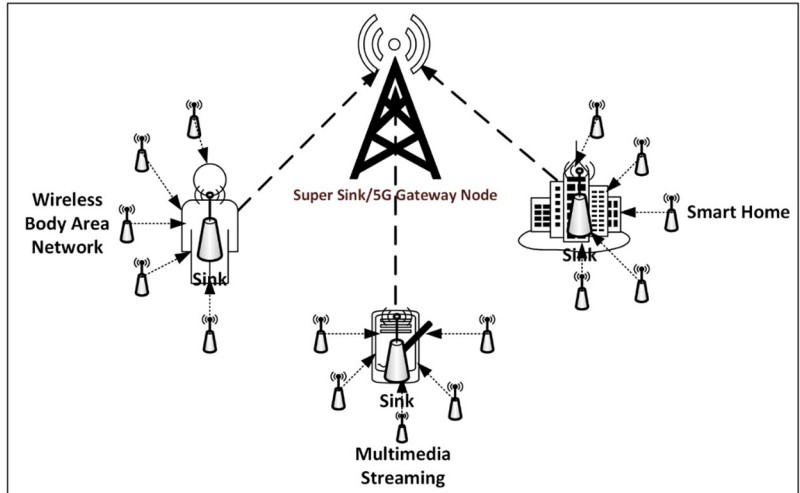

**Fig 1. RPL routing architecture and applications of WSN.**

the sink; however, the formation of DODAG does not take into consideration the heterogeneous nature of data and their priority status. This inadequacy makes it difficult for nodes that need urgent attention to be treated with the urgency they deserve as they constantly contend with nodes carrying delay-tolerant data. This practice encourages high latency and unnecessary power consumption which encourages suboptimal network lifetime. To improve network lifetime, the authors in [19–21] proposed a reduction in the number of data traversing the network by adopting data aggregation techniques. However, their suggestions did not adequately address the heterogeneous data nature of WSN as well as the varying priorities conveyed by these data.

5G, the new network paradigm for mobile communication is beginning to gain momentum [22]. While 5G is a relatively new technology, whose development started far back in 2013, standardized in 2016, and was practically rolled out for the first time in 2019 in South Korea [23], its major advantages are high bandwidth, very low latency, high speed, and support for network scalability [24], however, it has the following downsides, which are, high initial cost of implementation, high-power consumption, and low battery life for devices. These downsides can be compensated for by the collaborative implementation with WSN in a multi-tiered network architecture. WSN has the advantage of low-power consumption, which supports long battery life (lasting several months and years without a recharge) and miniaturized form factor, which make it robust and flexible for installation almost anywhere, and left unattended for a long duration. The implementation of 5G in a multi-tiered network with the proposed routing mechanism is presented in Section 3 of this article. While 5G is the promised network of the future, nevertheless it will not immediately extinct existing technologies such as WiMAX, 4G, and WSN, but work in conjunction with existing networks before taking the center stage when it reaches maturity in the near future. This article proposes and presents a routing mechanism that improves the routing of heterogeneous data across a multi-tier 4G and 5G network, with enhanced network performance.

The significant objective, method, and main contribution of this paper is detailed as follow:

- We proposed an improved and energy-efficient data routing mechanism called the Prioritized Shortest Path Computation Mechanism (PSPCM). The mechanism takes into consideration the heterogeneous nature of data traversing the WSN, hence, data were prioritized

according to the order of their importance using the Class of Service (CS) approach [5]. The heterogeneous data were classified into two major classes; High Priority Data (HPD), which is assigned a CS of 1, and Low Priority Data (LPD), which is assigned a CS of zero (0).

- Consequently, the proposed mechanism adopted an improved variant of the Dijkstra algorithm to generate energy-efficient routing paths for data guided by the assigned class of service of the data. The details of how the mechanism works are presented in Section 3.

- The advantage of the proposed mechanism is in its ability to treat heterogeneous data heterogeneously rather than homogeneously, thereby generating the shortest routing paths for each class of data from the source node to the sink while maintaining the integrity of the network. This is demonstrated by the results detailed in Section 4.

- Finally, from the outcome of the performance evaluation presented in section 4, in comparison with related mechanisms; ETX [3], QCOF [25], QWL-RPL [2], and RPL-EC and RPL-FL [26], PSPCM showed improved performance in packet delivery ratio (PDR), throughput, power consumption and convergence time, and better handling capabilities for heterogeneous data, without compromising the network integrity.

The research article is organized and summarized as follows; related work is presented in Section 2, which is followed by the proposed mechanism—prioritized shortest path computation mechanism in Section 3. Section 4 discusses the performance evaluation, while Section 5 presents results and discussion. The paper ends with a conclusion and future work in Section 6.

## 2. Related work

In the literature, researchers focus on the improvement of the RPL objective function, with little or no attention paid to resolving data priority issues. Without prioritization, data streaming through WSN has little or no meaning as it becomes difficult for critical data that are time-sensitive to be given the timely attention they deserve. With prioritization, child nodes are well guided on parent node selection and the path to route data through.

Musaddiq A. et al. in [2] presented queue and workload-based RPL (QWL-RPL). The mechanism approaches heterogeneity with respect to traffic patterns by considering the workload on a link and using the packet queue of a node to decide the congestion status of a node. It further approaches load imbalance problems in heterogeneous traffic by considering a network with two different types of traffic; the first is a network where the node transmits heterogeneous traffic, while the second is where traffic is generated at varying times. Results show that QWL-RPL outperformed compared mechanisms. However, the concept of heterogeneity is approached in the context of traffic patterns and the congestion status of the node rather than types of data (real-time or non-real-time) streaming through the network.

Lamaazi H. and Benamar N. in [3] presented an extensive survey on existing Objective Functions (OFs). RPL uses OFs to build its DODAG which is based on a set of rules which are dependent on either node or link metrics. It is the responsibility of the OF to ensure that the best child-parent paths, as well as optimal routing paths leading to the destination node, are selected when routing data across the network. The survey presented the ETX as the most optimal OF amongst other OFs. However, all OFs considered, including ETX does not take into consideration the heterogeneous nature of data traversing the network, nor the priorities these data convey, as these factors to a large extent impact the operational lifetime of the sensor network.

A reliable tree-based data aggregation method was proposed by the authors in [21]. In this scheme, for effective data aggregation, sensor nodes are arranged in the form of a binary tree

structure, while aggregated data are authenticated by a shared key. For error detection and correction, the scheme uses the dynamic generator polynomial-size for cycle error detection (CRC), which detects an error in a hop by hop manner, and in the event of an error, the packet is retransmitted by the previous node. To reduce the number and size of packet retransmission, data aggregation is performed by either summation or averaging. The approach helps to conserve the energy consumption in the network, which in turn improves the overall network lifetime. Results showed that the proposed scheme outperformed related schemes, however, the approach does not consider the heterogeneous nature of data traversing the network, subsequently the importance of data, which is represented by the properties conveyed by the data is not put into consideration.

Quality of Service and Congestion Aware Objective Function (QCOF) was proposed by Yousra, B.A et al. in [25]. This mechanism implements data prioritization using the following prioritized constraints; real-time constraint, energy constraint, link congestion constraint, and receptions constraint. Data is propagated according to data priority and if all the above conditions fail to yield any feasible route, then a new control message called New DODAG Request (NDR) message is created, this new control message creates an entirely new sink beside the original unreachable sink. QCOF uses the combination of node rank, energy constraint, time constraint, data priority, and congestion metrics to generate an optimized routing path towards the sink. Results show that QCOF outperforms OF0 and MRHOF. However, the newly created sink may not have the resource strength to handle the task of operating as a sink. Additionally, the dynamics of the prioritization mechanism are not clearly explained, which leaves room for questioning the validity of the result concerning its claimed consideration for data prioritization.

Energy utilization was identified as one of the major challenges of nodes in WSN, as this impacts the longevity of the sensor network. For optimal energy utilization, the author in [27] proposed an energy-efficient routing mechanism for prolonged operational network lifetime. The proposed scheme adopts a two-level routing approach, which improved packet delivery rate and improve energy consumption. That is, grouping the nodes into clusters and selecting a cluster head (CH), and selecting backup cluster heads (BCH). Nodes are made to cluster based on their residual energy, distance, and centrality, and secondly, each cluster is divided into four clusters such that data traversal is made through the most optimal CHs, and this is done with considers to node residual energy, relative distance, and centrality. In comparison with related schemes, this approach performed the most optimal, however, the approach did not take into cognizance the heterogeneous nature of data, nor the priorities conveyed by the data packets.

RPL-EC and RPL-FL were presented by Lamaazi H. and Nabil B. in [26]. RPL-EC approached path computation based on ETX and energy consumption while RPL-FL does the same with the techniques of fuzzy logic and the standard RPL trickle timer algorithm. These improved objective functions implementation was targeted at handling different WSN scenarios, which makes it difficult to account for a network with heterogeneous data under a single WSN. The performance of RPL-EC and RPL-FL were compared to the standard RPL and results show improved performance concerning network convergence time, packet delivery ratio, and end-to-end delays. However, the issue of data prioritization for heterogeneous data is not considered which makes the mechanism not applicable for heterogeneous WSN scenarios.

As designed, and owning to its small form-factor, the limited energy of the sensor node was identified by the authors in [19] as one of the major factors contributing to its early energy depletion, hence an optimal data aggregation mechanism is needed to improve the energy utilization of nodes in WSN for a prolonged operational lifetime. The authors suggested that

reducing the number of data packets transmitted across the network will help reduce the energy consumption, hence they proposed the Secured Hybrid Structure Data Aggregation (SHSDA) mechanism. In this mechanism, each child node is assigned a parent node to propagate its data through. To improve data security, lightweight symmetric encryption is adopted, and a key is distributed across child and parent nodes for secured data transactions. Data is sent from the child node to the parent node until it reaches the root node through a star topological structure before the data finally arrive at the base station (BS) through a tree topology. From the results, SHSDA outperformed related schemes, however, it failed to consider the heterogeneous nature of data and the priority conveyed by the data across the network as these factors a significantly important to affect the longevity of the sensor network.

The Load-balanced and Energy-aware Opportunistic Routing with adaptive duty cycling in multi-channel WSNs (LEOR) were presented by [28]. The scheme proposed an energy-efficient duty cycling and opportunistic routing mechanism. LEOR supports scaling networks with inter-node and inter-channel level load balancing. It utilized channel prioritization based on channel utilization to determine the number of busy nodes and to avoid network interference by other nodes. To increase route quality, opportunistic routing is used to restrict data forwarding to already busy nodes which result in energy wastage and data retransmission. This results in improved end-to-end delays and packet delivery ratio. However, LEOR does not consider the heterogeneous data nature of WSN, this results in an inappropriate representation of data streaming through the network, and suboptimal network performances.

In [29], the authors identified unbalanced energy consumption in data routing across the network as one of the major issues in WSN. To alleviate the issue, the authors proposed a Multipath Routing Mechanism for homogeneous WSN. This mechanism works in three ways; clustering nodes, discovering optimal routing paths between cluster heads, and keeping the routing paths maintained. The scheme adopts the firefly algorithm for node clustering, while fuzzy logic is used to route data between cluster heads. This approach results in the creation of two routing paths; the primary path, used for normal data routing, and the backup paths used when routing through the primary paths fail to deliver data to the BS. Finally, in the event of route failure such as route breakage, route maintenance, the mechanism restarts the whole route to re-establish the primary and backup routes. From the results, the proposed mechanism outperformed related schemes, however, the authors failed to take into consideration the heterogeneous nature of data traversing the network as well as the priorities conveyed by the data as this significantly affects the overall performance of the sensor network.

The Energy Efficient Correlation Aware hop-by-hop opportunistic Routing protocol (EECAR) was proposed by [30]. The protocol approached the improvement of data routing by careful improvement of the opportunistic routing protocol which consists of lossy links with the implementation of greedy information sensor selection algorithm, which greedily select the appropriate next-hop node to improve network efficiency without compromising the network integrity. EECAR takes into consideration the following factors; spatial correlation characteristics among sensor readings, broadcast nature of wireless channels, and the residual battery energy of nodes. Results show that EECAR performed better than related schemes, however, the heterogeneous data nature of WSN was not taken into consideration, this limitation undermine the network performance.

A secured data aggregation scheme was proposed in [20] to reduce the excessive energy consumption of nodes in WSN. This scheme works in three stages; intra-cluster aggregation; which is responsible for data routing with a cluster; inter-cluster aggregation, which is responsible for data routing between different clusters, and finally data transmission, which handles successful data routing between different clusters. Fuzzy logic is used for intra-cluster aggregation, while for inter-cluster aggregation, tree topology is created between cluster head nodes.

This aggregation used the dragonfly algorithm. Finally, for data transmission, the columnar transportation cipher is used to securely transfer data across all cluster members and cluster heads. This scheme shows the most optimal performance in comparison with related schemes, however, it failed to consider the heterogeneous nature of data transmitted across the network alongside the priorities these data convey.

## 2.1. Problem statement

The literature reveals that the current standard and mechanisms proffered in the literature do not provide a workable and sustainable prioritization mechanism to categorize the heterogeneous WSN data stream into various priority classes [2]. To resolve this issue, the routing protocol should be improved to handle and perform data routing of heterogeneous data with varying data priorities. Without data prioritization, node ranking, child-parent node selection, and preferred routing path to the sink will be decided by a single or same objective function instead of a different objective function that is unique to the priority of data conveyed by the node [6]. Subsequently, such data will have to contend with delay-tolerant data in the sensor network. This practice could be detrimental especially in instances where the data is time-bound and mission-critical. This consequently leads to increase packet collision, high latency, reduced throughput, poor packet delivery ratio, and high-power consumption which eventually shortens the operational lifetime of the already power-constrained nodes. Mitigating against these inadequacies, the priority of data should be used as an objective function to decide how the data is routed from source to destination nodes, as such objective function will be optimized to meet the need of the data priority thereby yielding an optimized shortest and energy efficient path to the sink, which improves the overall network lifetime.

## 3. The proposed mechanism—Prioritized Shortest Path Computation Mechanism (PSPCM)

This section and beyond discusses the contribution made to improve data routing in the network layers of the WSN protocol stack. The improved mechanism considers the priorities inherent in the data streaming through the network, which represents and places importance on the heterogeneity of data traversing the WSN. The flowchart of the research methodology for the PSPCM mechanism is shown in Fig 2.

### 3.1. Design and implementation of PSPCM—Data classification and prioritization

The application of PSPCM is presented in multi-tiered network architecture, where tier 1 represents the 5G domain, and tier 2 represents the WSN domain. Furthermore, a zoomed-in section of Fig 3 for smart home application with the proposed mechanism—PSPCM is presented in Fig 4. At tier 1, data is routed across the WSN using the proposed mechanism—PSPCM, while tier 2 handles the propagation of data through the 5G internet cloud for either storage or instantaneous processing and utilization.

To comprehend the contributions made in this research, a smart home is considered as shown in Fig 4. Pre-configured sensor nodes are attached to various parts of the home and sensed parameters are sent to the homeowner who may be at a local or remote location through the internet to his/her handheld device or other convenient internet-enabled monitoring devices. Two types of nodes are installed in the setup; High Priority Node (HPN)—which generates High Priority Data (HPD), installed in locations that generate time-sensitive and critical data, and Low Priority Node (LPN)—which generates Low Priority Data (LPD),

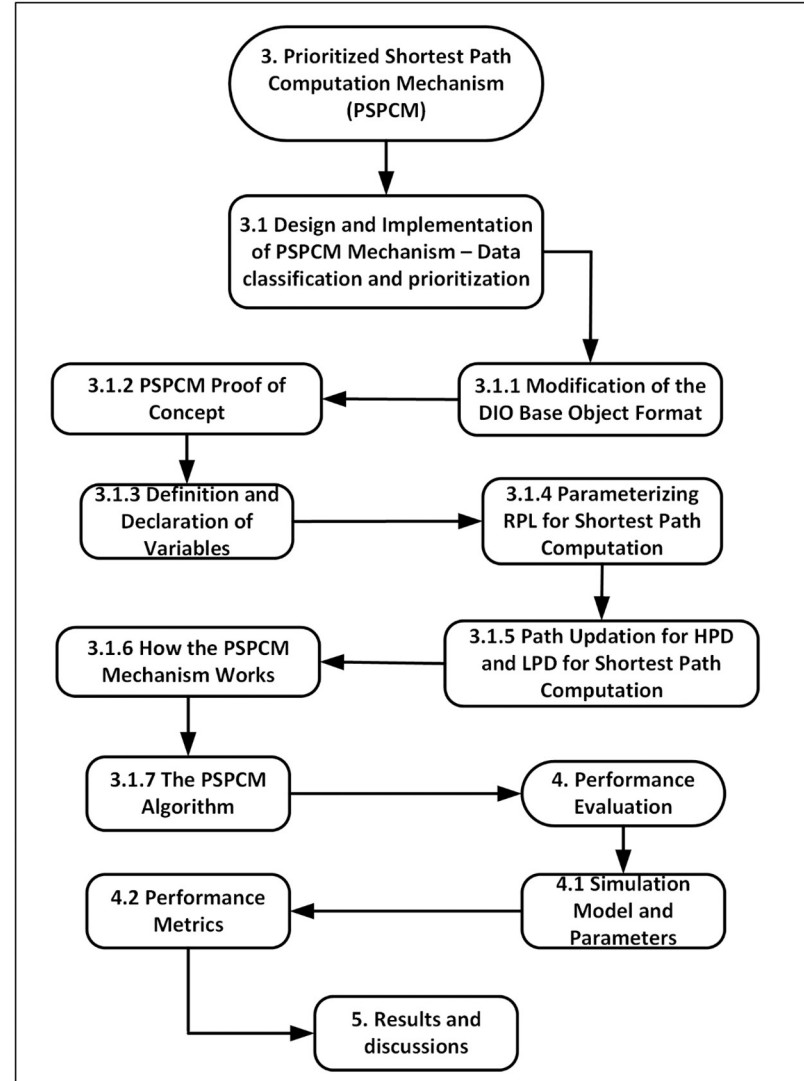

**Fig 2. PSPCM methodological approach.**

installed in locations that generate delay-tolerant and less-critical data. The heterogeneous data (HPD and LPD) generated by the smart home network are classified and prioritized into two (2) major classes, using the techniques of *Class of Service (CS)* [5]. One of the objectives of this research is to ensure that data are not only successfully sent to the destination node, but also through the shortest and energy-efficient path. To achieve this aim, the prioritized shortest path computation mechanism was devised, which solves the problem of single-source shortest path (SSSP) was used to establish an optimized, power-efficient, and load-balanced routing path leading to the sink for all source nodes in the network. To effectively capture the prioritized data, the RPL DIO message base object was modified to accommodate data of varying priorities. These modifications yielded the goal of this research which is the *prioritized shortest path computation mechanism (PSPCM)*.

In this mechanism, data traverses across the network in bytes. To ensure that the proposed mechanism—PSPCM operates in the most energy-conservative manner, IP Header

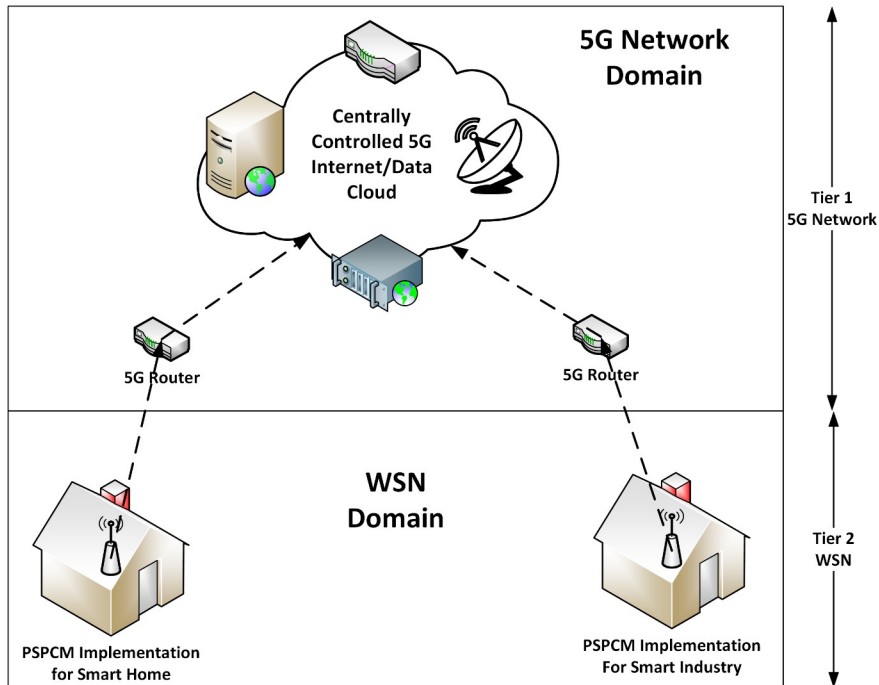

**Fig 3. PSPCM Implementation with WSN and 5G in a multi-tiered network architecture.**

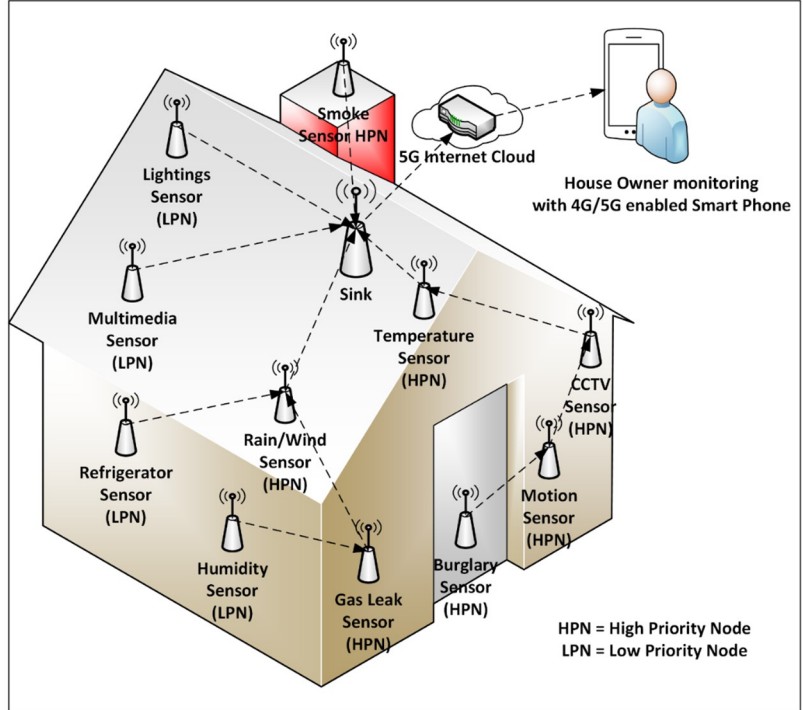

**Fig 4. Application of PSPCM in smart home—Zoomed-in.**

| Frame Header (25 Bytes) | Link-layer Security (21 Bytes) | IPv6 Header (40 Bytes) | UDP Header (8 Bytes) | Payload (Variable) (33 Bytes) |
|---|---|---|---|---|

**Fig 5. IEEE 802.15.4 MAC frame headers and payload field.**

Compression (IPHC) techniques—LOWPAN_IPHC and LOWPAN_NHC, were used to compress the IPv6 header as well as the UDP header. The maximum transmission unit (MTU) of the IEEE 802.15.4 is 127 bytes of which 25 bytes is used for the MAC frame header, which leaves the payload with 102 bytes [31]. Of these 102 bytes, link-layer security (LLSEC) uses 21 bytes, IPv6 header uses 40 bytes and user datagram protocol (UDP) header uses 8 bytes. This eventually leaves the payload with only 33 bytes. Fig 5 shows the IEEE 802.15.4 MAC frame header format and the payload field.

The 33 bytes (variable) may sometimes not be sufficient to accommodate the payload, hence the need to increase the size of the payload capacity. To achieve this, the IP and UDP headers are compressed from 48 bytes to 6 bytes, this eventually increases the payload size to 75 bytes which are variable, depending on the size of the sensed data [32]. Fig 6 shows the IEEE 802.15.4 IPv6 and datagram header compression. To conserve energy, while ensuring optimal performance of the proposed mechanism, a payload size of 40 bytes is used in our experimentation.

**3.1.1. Modification of the DIO base object format.** The WSN protocol stack runs in all nodes (that is, HPN and LPN), the network layer, which is responsible for data routing, accommodates the RPL *DIO Base Object*. The unused *Reserved* field of the DIO base object which holds 8 bits is modified to accommodate data priority classes as shown in Fig 7(a) and 7(b) indicated by the arrow. The proposed mechanism uses only two CS tags; CS = 0, for bits 00000000, and CS = 1, for bits 00000001. CS = 0 represents Low Priority Data (LPD) which is delay-tolerant and non-time-bound for non-real-time data, (this CS value is assigned into the LPN), while CS = 1 represents High Priority Data (HPD) which is time-sensitive for real-time data, (this CS value is also assigned into the HPN). In order words, HPN is a pre-configured node with CS = 1, an LPN is a pre-configured node with CS = 0. The CS tags serve as a *deciding factor* for how the proposed PSPCM routing mechanism operates. Table 1 shows the data prioritization classification and the class of service tags assigned to each node.

**3.1.2. PSPCM proof of concept.** While bearing in mind the heterogeneous data nature of the WSN, the DODAG graph generated by the RPL network is a directed graph that exhibits a single source shortest path problem (SSSP). To solve this problem, the improved derivate of the Dijkstra algorithm was adopted [33], which is famous for solving a graph of single-source shortest path problems. It works by finding the shortest path for weighted (path cost) and

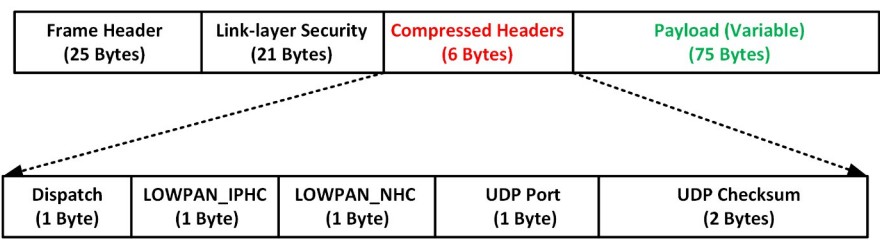

**Fig 6. IEEE 802.15.4 IPv6 and datagram header compression.**

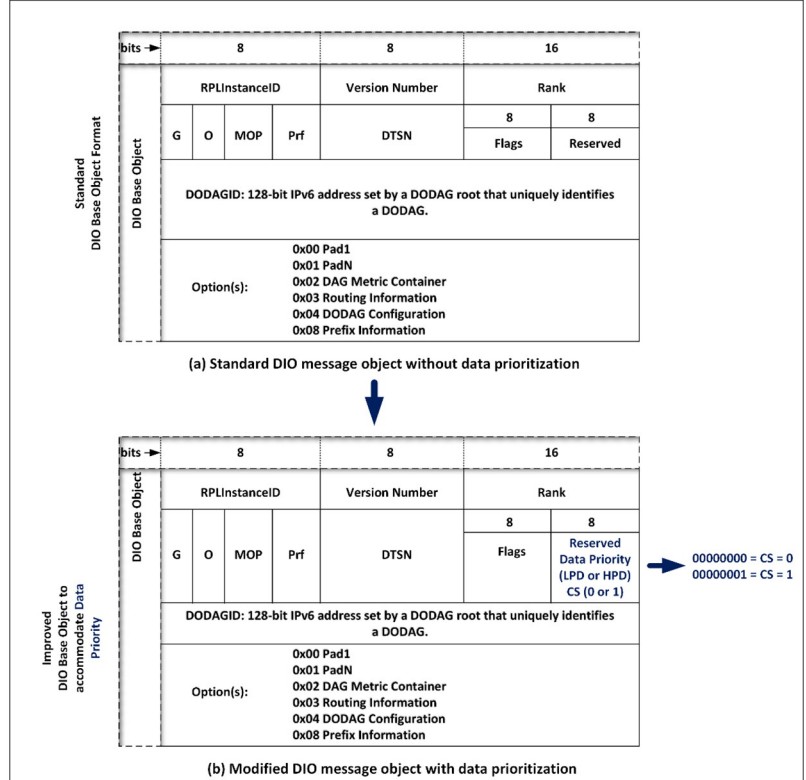

**Fig 7. (a) Standard DIO base object (b) Modified DIO base object.**

unweighted (no path cost) or directed (with directional arrow) and non-directed (without a directional arrow) multi-path graph as is the case for DODAG, by routing data packets from source node to destination node. With the priority of the data packet, which is determined by the data CS, PSPCM finds the *least path cost* and *shortest connecting distance* between two nodes in the presence of multiple routes connecting both nodes. The following conditions must be satisfied to be able to compute the prioritized shortest path:

- the graph (virtual data propagation map) must contain nodes connected by weighted links with path cost.

- the values of the path cost also called weighted edges must be non-negative.

- the graph must contain both directed and or non-directed graphs.

- the graph must be a single-source, single-destination shortest path problem.

Consider a graph $G = (V, E)$, where $V$ is a set of finite nodes; $V = \{u_1, u_2, \ldots \ldots, v\}$ and $E$ is a set of connecting edges also referred to as weights or path costs $E = \{w_1, w_2, \ldots \ldots, w_n\}$. The

**Table 1. Data prioritization using the class of service technique.**

| Priority Status | Class of Service (CS) | Event Type |
|---|---|---|
| High Priority Data (HPD) | 1(Decimal value) | Real-time |
| Low Priority Data (LPD) | 0 (Decimal value) | Non-Real-time |

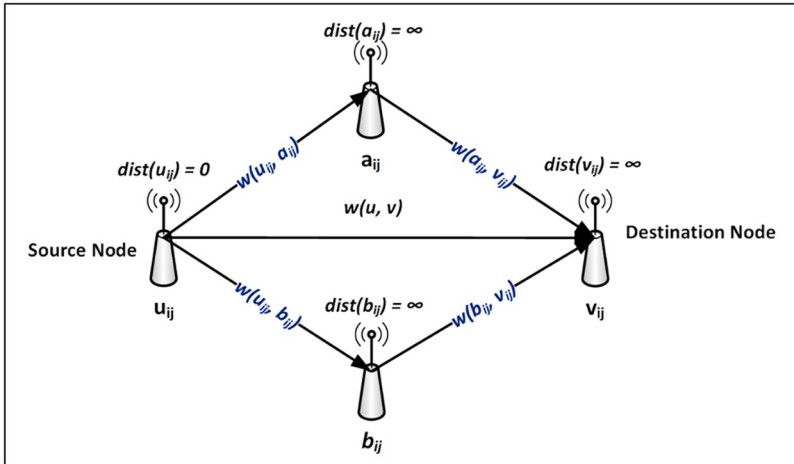

**Fig 8. PSPCM shortest path computation.**

nodes $u_{ij}$ and $v_{ij}$ are the source and destination nodes respectively, with weight $\overrightarrow{w\left(u_{ij} v_{ij}\right)}$, and the initial node of an edge is the terminal node of the preceding edge in the sequence. The path cost from the source node $n_1$ to the destination node $n_n$, is the sum of all edges from source to destination node, assuming that the network consists of a directed path from the source node to every other node in the network. See Fig 8.

**3.1.3. Definition and declaration of variables.** Three variables are initialized as follow;

- *dist*; represents distance. The distance between the source and itself is initialized as $dist(u_{ij}) = 0$, and the distance between source to other nodes with the unknown *dist* is given as infinity; $dist(v_{ij}) = \infty$. This is done at the beginning of the algorithm run, as the distance between the source node to adjacent nodes will be recomputed and updated as the graph is traversed until the destination node is reached.

- *Q*; represents the Queue, which holds all the unvisited nodes in the graph. Once all nodes are visited, and the shortest path is established, the *Q* will be empty.

- *S*; is an empty set at the beginning of the algorithm run, which holds the set of visited nodes. At the end of the algorithm run, *S* will contain all visited nodes.

  The steps of how the algorithm works are detailed below:

**Step 1**: While *Q* is not empty, pop the node $v_{ij}$ that is not already in *S* from *Q* with the smallest $dist(v_{ij})$. In the first run, the source node $u_{ij}$ will be chosen because $dist(u_{ij})$ is initialized to be zero (0). In the next run, the next node with the smallest distance will be selected.

**Step 2**: Add the node $v_{ij}$ to the list of visited nodes *S*, indicating that it has been visited.

**Step 3**: Update ***dist*** values of adjacent nodes of the current node $\boldsymbol{u_{ij}}$ as follows: for each new adjacent node $\boldsymbol{v_{ij}}$, that is, if; see Eq 1,

$$dist\left(u_{ij}\right) + \overrightarrow{weight\left(u_{ij}, v_{ij}\right)} < dist\left(v_{ij}\right) \tag{1}$$

and there is a minimum distance found for $v_{ij}$, then update $dist(v_{ij})$ to; see Eq 2,

$$dist\left(v_{ij}\right) = dist\left(u_{ij}\right) + \overrightarrow{weight\left(u_{ij}, v_{ij}\right)} \tag{2}$$

else do not update $dist(v_{ij})$.

**Step 4**: Finally, the shortest distance $SD_{min}$, from source to destination node can be estimated by; see Eq 3,

$$SD\ min \sum_{i=1}^{n} dist(v_n)_{min} \tag{3}$$

All nodes in the graph have been visited and the shortest distance from the source to the destination node is established and estimated by Eq 3, where $dist(u_{ij})$ are a scalar quantity and weight or path cost $\overrightarrow{w\left(u_{ij} v_{ij}\right)}$ is a vector quantity. This approach is implemented by adopting the Fibonacci Heap with minimum priority queueing [34] and adjacency list data structure with time complexity shown by Eq 4.

$$Time\ complexity\ (Tc) = O[|E| + |V|Log|V|] \tag{4}$$

Where $|E|$ represents the number of edges and $|V|$ represents the number of nodes.

**3.1.4. Parameterizing RPL for shortest path computation.** To implement the prioritized shortest path computation mechanism in RPL, it is necessary to identify and parameterize relevant parameters that satisfy the algorithm. Node and link metrics are identified and each node possesses at least three of these following parameters:

- **Link ETX**: indicates the path cost between two nodes [3].

- **Node rank (R)**: the position of the node relative to neighboring nodes and the sink node [2].

- **Node energy (NE)**: remaining energy resident in a node [9]

Fig 9 shows a multi-hop PSPCM routing path computation architecture and topology, which generates heterogeneous data of HPD and LPD. The weight or path cost between two adjacent nodes is represented as; $ETX_{ij}$, $HC_{ij}$. The value of ETX is proportional to the inverse of PDR and proportional to the transmission (Tx) range. That is, ETX increases with increasing Tx range until the maximum Tx range is reached, which is the boundary between the Tx range and interference range [27]. See Eqs 5 and 6, while HC is represented by one (1) (each hop has a value of 1) [18, 35, 36]. Each node has an instantaneous node rank $R$ and residual energy $RE$. ETX is used as an objective function to generate the routing path for HPD, while HC is used as an objective function to generate a routing path for LPD.

$$ETX = \frac{1}{PDR}\alpha Tx_{range} \tag{5}$$

Eq 5 can be further written as Eq 6.

$$ETX = \frac{1}{d_f \times d_r} = \tau Tx_{range} \tag{6}$$

Where $d_f$ is the forward delivery ratio, which is the probability that the packet is received at the destination node, $d_r$ is the reverse delivery ratio, which is the probability that the ACK packet

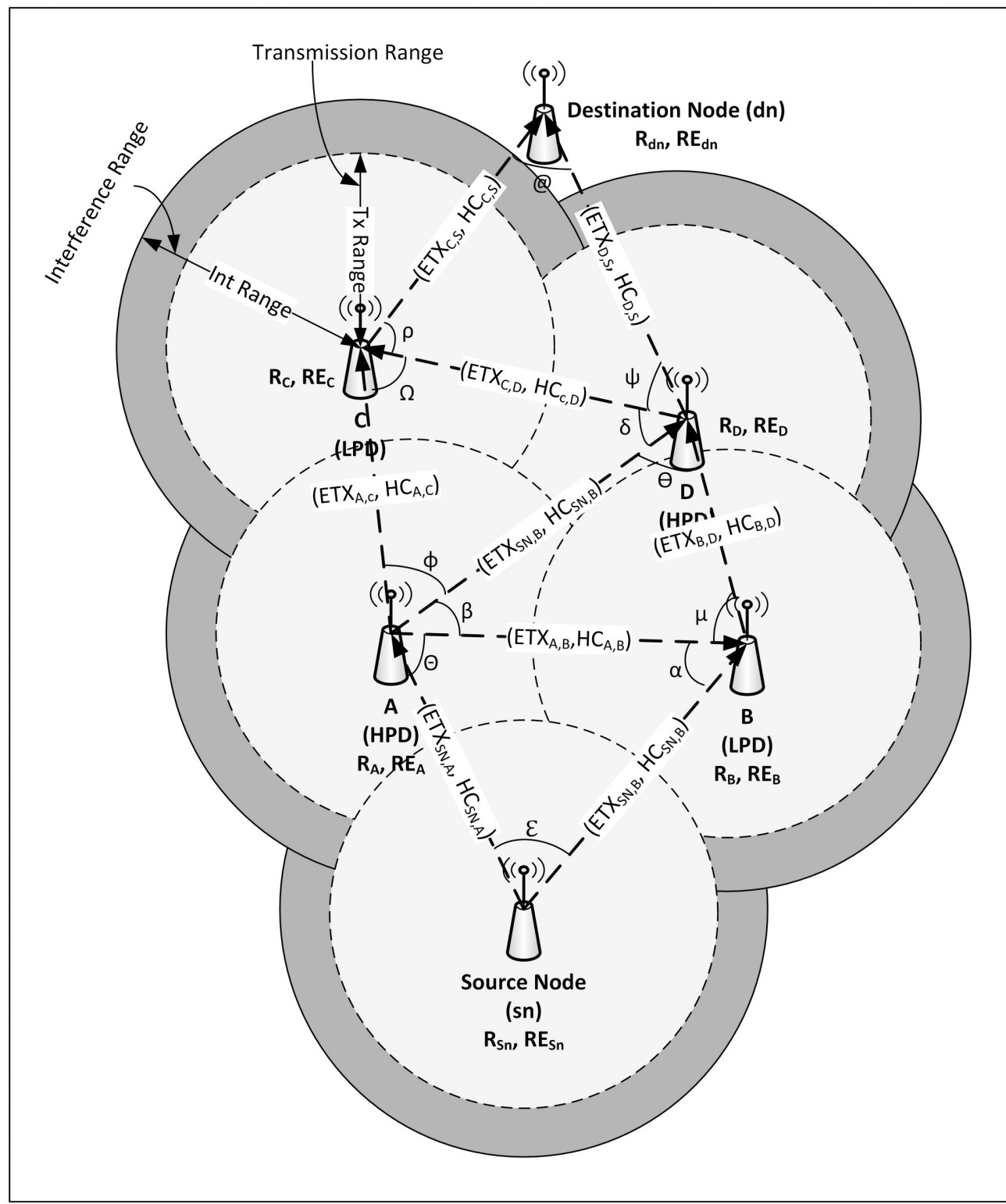

**Fig 9. PSPCM routing path computation architecture and topology.**

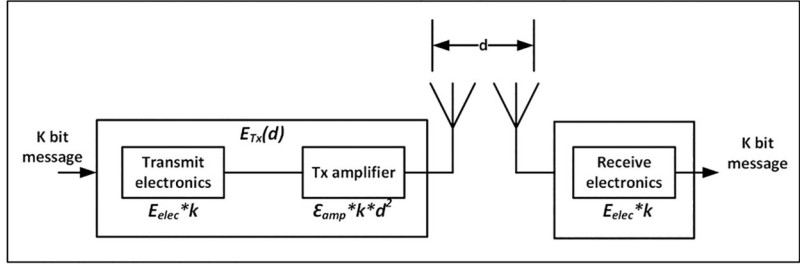

**Fig 10. The Heinzelman first order radio model.**

is delivered or received by the sender node, and $\tau$ is the *transmission* or *propagation constant* [37, 38]. Subsequently, the node energy represented by Eq 7 can be estimated by the Heinzelman first order radio model [39], see Fig 10.

$$E_{Total} = E_{TX}(k, d) + E_{RX}(k, d) \tag{7}$$

Where, $E_{TX}(k, d)$ represents the energy expended per bit by the transmitter, and $E_{RX}(k, d)$ represents the energy expended per bit by the receiver circuit, see Eqs 8 and 9. $k$ represents the number of bits in the message, $E_{elec}$ is the basic power dissipation coefficient of the transceiver circuit. $\in_{fs}$ and $\in_{amp}$ are the energy consumption for free space and multipath channel model power amplifier respectively and $d$ is the node distance from the sink. When $d < d_0$, the algorithm adopts the free space model with $d_2$, otherwise $d_4$ is used for the multipath fading channel model.

$$
\begin{aligned}
E_{TX}(k, d) \quad &= E_{elec} * k + \in_{fs} * k * d^2, \quad if \ d < d_0 \\
&= E_{elec} * k + \in_{amp} * k * d^4, \quad if \ d \leq d_0
\end{aligned}
\tag{8}
$$

$$E_{RX}(k, d) = k * E_{elec} \tag{9}$$

**3.1.5. Path updation for HPD and LPD for shortest path computation.** Considering Fig 9, for node *sn* (source node) with adjacent nodes *A* and *B*, the distance between nodes *sn* and *A* is represented by $\overrightarrow{sn, A}$, nodes *sn* and *B* by $\overrightarrow{sn, B}$, and nodes *A* and *B* by $\overrightarrow{A, B}$, respectively. Subsequently, the angular separation between the distances $\overrightarrow{sn, A}$, and $\overrightarrow{sn, B}$ is given as $\mathcal{E}°$, between $\overrightarrow{sn, A}$ and $\overrightarrow{A, B}$, as $\Theta°$, and $\overrightarrow{A, B}$ and $\overrightarrow{sn, B}$ as $\alpha°$ respectively.

The distance $\overrightarrow{sn, A}$ can be estimated by Eq 10:

$$\overrightarrow{sn, A} = \sqrt{\left[ \overrightarrow{(A, B)}^2 + \overrightarrow{(B, sn)}^2 \right] - 2 \left( \overrightarrow{A, B} \right) \left( \overrightarrow{B, sn} \right) Cos\alpha} \tag{10}$$

Similarly, the distance $\overrightarrow{A, B}$, can be estimated by Eq 11:

$$\overrightarrow{A, B} = \sqrt{\left[ \overrightarrow{(B, sn)}^2 + \overrightarrow{(sn, A)}^2 \right] - 2 \left( \overrightarrow{B, sn} \right) \left( \overrightarrow{sn, A} \right) Cos\mathcal{E}} \tag{11}$$

and the distance $\overrightarrow{B, sn}$, can be estimated by Eq 12:

$$\overrightarrow{B, sn} = \sqrt{[(\overrightarrow{sn, A})^2 + \overrightarrow{(A, B)}^2] - 2\left(\overrightarrow{B, sn}\right)\left(\overrightarrow{sn, A}\right)Cos\Theta} \tag{12}$$

where distances $\overrightarrow{sn, B} = \overrightarrow{B, sn}$, $\overrightarrow{sn, A} = \overrightarrow{A, sn}$, and $\overrightarrow{A, B} = \overrightarrow{B, A}$

Referencing Eqs 1 and 2, and considering Fig 9, path updation for the path $\overrightarrow{sn, A}$ can be estimated by Eqs 13 and 14 for HPD and LPD respectively.

$$HPD_{dist}(A) = dist(sn) + w(sn, A), \ if \ dist(A) > dist(sn) + w(sn, A) \tag{13}$$

$$LPD_{dist}(A) = dist(sn) + w(sn, A), \ if \ dist(A) > dist(sn) + w(sn, A) \tag{14}$$

Subsequently, path updation across nodes A, C to destination node $dn$, and the paths of other nodes leading to the destination node can be obtained in like manner as Eqs 13 and 14. Substituting the RPL parameterized variables ($R$, $RE$, $ETX$, and $HC$) as shown by Eqs 15 and 16 into Eqs 13 and 14, we obtain Eqs 17 and 18 for HPD and LPD respectively.

$$dist(sn) = R + RE, for \ HDP, and \ dist(sn) = R \ for \ LPD \tag{15}$$

and

$$w(sn, A) = \begin{cases} ETX, \ for \ HPD \\ HC, \ for \ LPD \end{cases} \tag{16}$$

Path updation for adjacent next-hop-node ($nhn$) for HPD and LPD respectively is;

$$HPD_{dist}(nhn) = R + RE + ETX(sn, nhn), for \ HPD_{dist}(nhn) > R + RE + ETX(sn, nhn) \tag{17}$$

$$LPD_{dist}(nhn) = R + HC(sn, nhn), for \ LPD_{dist}(nhn) > R + HC(sn, nhn) \tag{18}$$

Where ETX and HC are equivalent to the distance between two nodes for HPD and LPD. Let the *nhn* from the *sn* be node *A*, then, Eqs 17 and 18 and can be rewritten and this gives Eqs 19 and 20

$$HPD_{dist}(A) = R + RE + ETX(sn, A), for \ HPD_{dist}(A) > R + RE + ETX(sn, A) \tag{19}$$

$$LPD_{dist}(A) = R + HC(sn, A), for \ LPD_{dist}(A) > R + HC(sn, A) \tag{20}$$

where $ETX(sn, A) \Leftrightarrow \overrightarrow{sn, A}$, and $HC(sn, A) \Leftrightarrow \overrightarrow{sn, A}$, and substituting the values of $\overrightarrow{sn, A}$ from Eq 10 into Eqs 19 and 20, we have Eqs 21 and 22

$$HPD_{dist}(A) = R + RE + \sqrt{[\overrightarrow{(A, B)}^2 + \overrightarrow{(B, sn)}^2] - 2\left(\overrightarrow{A, B}\right)\left(\overrightarrow{B, sn}\right)Cos\alpha}, for \ HPD_{dist}(A)$$
$$> R + RE + ETX(sn, A) \tag{21}$$

$$LPD_{dist}(A) = R + \sqrt{[\overrightarrow{(A, B)}^2 + \overrightarrow{(B, sn)}^2] - 2\left(\overrightarrow{A, B}\right)\left(\overrightarrow{B, sn}\right)Cos\alpha}, for \ LPD_{dist}(A)$$
$$> R + HC(sn, A) \tag{22}$$

The shortest path from the source to the destination node is the path with the least weight/ minimum path cost across all paths. This can be estimated by Eqs 23 and 24. Where $HPD_{SP}(sn,$

*dn*) and $LPD_{SP}(sn, dn)$ represents the shortest path for HPD and LPD respectively, and *sn* is the source node, *dn* is the destination node, and *nhn* is the next-hop-node (adjacent node). Eqs 23 and 24 are known as the *Prioritized Shortest Path Computation Mechanism (PSPCM)* for HPD and LPD respectively.

$$HPD_{SP}(sn, dn) = min \sum_{i \neq j}^{n} HPD_{dist}(nhn) \qquad (23)$$

$$LPD_{SP}(sn, dn) = min \sum_{i \neq j}^{n} LPD_{dist}(nhn) \qquad (24)$$

**3.1.6. How the PSPCM mechanism works.** Having established how path computation is estimated for HPD and LPD, the goal of the PSPCM routing mechanism is to achieve an optimized shortest routing path that considers data priorities using the CS approach which gives meaning to data routed through the sensor network. The PSPCM decides how hop-to-hop child-parent nodes optimized routing paths to the sink are established. Once the network is powered on, all nodes including the sink boots up and undergoes initialization, after this, the sink broadcasts the DIS, which is a query-like control message to inquire of nodes rank (R), node energy (NE), class of service (CS), expected transmission cost (ETX) and hop count (HC). In response, nodes reply with DIO, which holds the requested information, and this is transmitted using the broadcast mode to the sink. Nodes with CS = 1 computes all possible paths leading to the sink with Eq 21 for HPD, and if CS = 0 with Eq 22. Once the DIO arrives at the sink, the sink computes the shortest path from all possible source nodes with Eq 23 for HPD or Eq 24 for LPD. The sink then replies to the source with DAO which holds the information of the shortest possible path leading to it (sink). At this point, the network converges and is ready to transmit actual data. However, these shortest paths are not utilized throughout the lifetime of the network, as all paths are periodically re-computed with the help of the trickle timer trigger, particularly in the event of network instability, possibly resulting from the node (s) running out of battery power, or changing rank. Route re-computation ensures that packets are routed through the most energy-efficient shortest path which supports long battery life and prolonged network lifetime.

The steps of how the mechanism works are described below which are followed by the flowchart, presented in Fig 11, and then the PSPCM algorithm.

Step 1:. Variable declaration and initialization

Step 2:. Sink sends DIS to nodes and nodes reply with modified DIO (contains; CS, R, NE, ETX, and HC)

Step 3:. If CS = 1, the source node computes all possible paths to sink with Eq 21 for HPD, else compute the path to sink with Eq 22 for LPD.

Step 4:. If DIO arrived at the sink, then move to Step 5, else move to Step 2.

Step 5:. If CS = 1, sink computes the shortest path to source node with Eq 23 for HPD, else computes the shortest path to source node with Eq 24 for LPD.

Step 6:. Sink selects the shortest path and sends information of the shortest path to the source node with DAO acknowledgment. The network converges and is ready to send actual data.

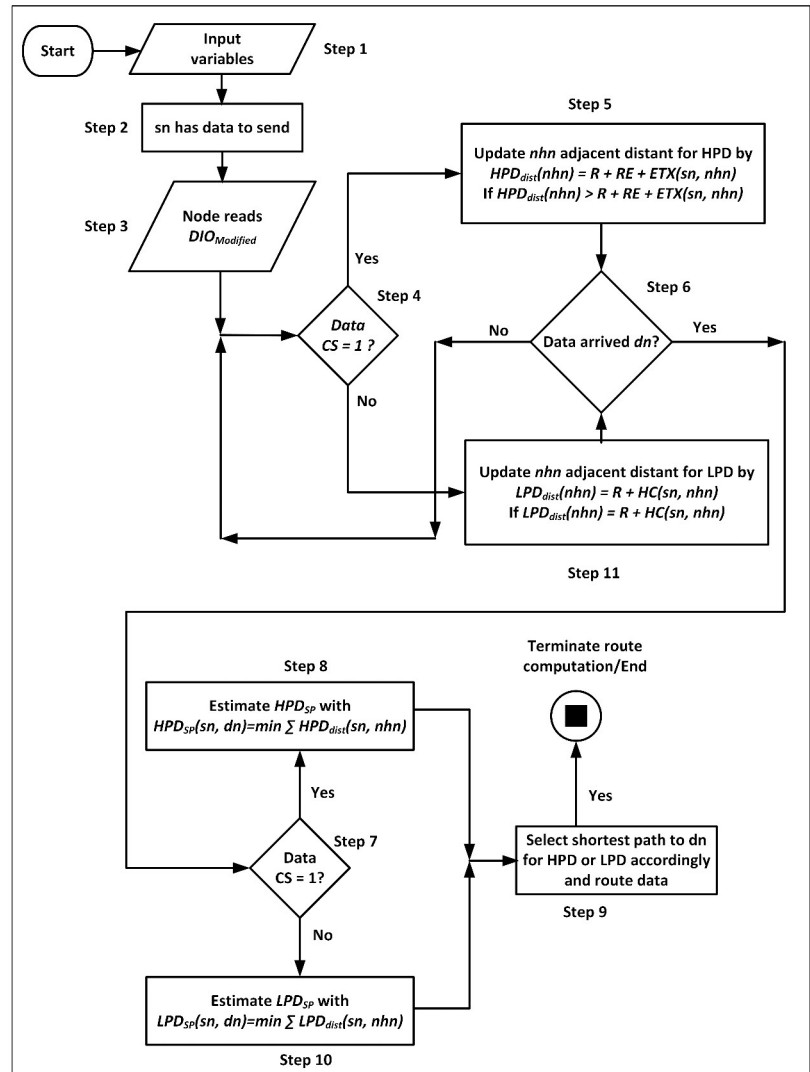

**Fig 11. Prioritized Shortest Path Computation Mechanism (PSPCM) flow chart.**

Step 7:.  Nodes send data to sink through the shortest possible path.

**3.1.7. The PSPCM algorithm.**   The algorithm of how PSPCM works are shown in Algorithm 1:

**Algorithm 1: SPPB-OF Algorithm**
**START**
**//Variable declaration**
1: **INITIALIZE:** $HC = 1$; $dist(sn) = 0$; $dist(nhn) = \infty$ //distance from source node to self is 0 and distance from source node to $nhn$ is infinity **[Step 1]**
2: Sink send $DIS$ to all nodes and nodes reply with $DIO_{Modified}$ (R, NE, CS, ETX, and HC) **[Step 2]**
4: **IF** CS = 1 **[Step 3]**   //For HPD

5:    **THEN** Node update *nhn* adjacent distance for HPD by $HPD_{dist}(nhn) = R + RE + ETX(sn, nhn)$, if $HPD_{dist}(nhn) > R + RE + ETX(sn, nhn)$
6:    **ELSEIF** Data arrived sink *(dn)* **[Step 4]**
7:      **IF** Data CS = 1 **[Step 5]**
8:        **THEN** Sink **e**stimates the shortest path to the source node for HPD with $HPD_{SP}(sn, dn) = \min \sum_{i \neq j}^{n} HPD_{dist}(sn, nhn)$ **[Step 6]**
9:        Sink select shortest path to the source node for HPD and send shortest path information to source node using DAO **[Step 7]**
10:       **ELSE** Sink estimate the shortest path to the source node for LPD with $LPD_{SP}(sn, dn) = \min \sum_{i \neq j}^{n} LPD_{dist}(sn, nhn)$ **[Step 6]**
11:       Sink select shortest path to the source node for LPD and send shortest path information to source node using DAO **[Step 7]**
12:     **ENDIF**
13:   **ELSE** Node moves to **Step 2**
14:   **ENDIF**
15: **ELSE** Node update *nhn* adjacent distance for LPD by $LPD_{dist}(nhn) = R + HC(sn, nhn)$, if $LPD_{dist}(nhn) > R + HC(sn, nhn)$ **[Step 3]**
16: **ENDIF**
**END**
**OUTPUT:** A energy-efficient and optimized shortest path is obtained for HPD or LPD respectively from the source node to the destination node, resulting in improved network performance with support for extended network lifetime.

## 4. Performance evaluation

The performance of PSPCM, with regards to related mechanisms, is carried out by performance evaluation through simulation, and the results of HPD and LPD were compared with respect to performance metrics (see Subsection 4.2). Furthermore, PSPCM was benchmarked against related data heterogeneity and priority-based mechanisms; ETX [3], QCOF [25], QWL-RPL [2], and RPL-EC and RPL-FL [26]. The simulation is performed with Cooja [2, 40]; a Java-based simulator built into the Contiki Operating System. Contiki OS is a standard operating system for IoT. Embedded in the simulator is the Zolertia Z1 mote (node) which utilizes the IEEE 802.15.4 standard. The mote uses the CC2420 radio transceiver and the MSP430 low-power microcontroller, operating in the 2.4GHz ISM frequency with a data rate of 250Kbps and guarantees maximum efficiency with low-power at low cost [41]. Each node transmits a payload size of 40 bytes, see Table 2. Performance comparison and benchmarking were performed on the following performance metrics; packet delivery ratio (PDR), power consumption, throughput, and convergence time, with details presented in Subsection 4.2.

### 4.1. Simulation model and parameters

Consider a multi-hop low-power and lossy network (LLN) setup which forms an RPL routing topology with data transmission from a child node to parent node and vice versa, creating a DODAG graph. The graph consists of nodes ranked incrementally from the sink to the child nodes. Consistent with the following simulation models [2, 6, 26]. Tree topology and random node placement were adopted with one sink and three hundred (300) nodes for HPD and LPD comparison; and two hundred (200) nodes for comparison with mechanisms in the literature, in a 300 by 300 square meter (m$^2$) sensor field with adjacent nodes having a clear line of sight to each other. Each simulation was executed for 2000 seconds and incrementing the number of nodes starting from 15 and 10 nodes up to a maximum of 300 nodes and 200 nodes for HPD/LPD comparison and benchmarking respectively. The simulation configuration settings and parameters are shown in Table 2 below.

**Table 2. Simulation configurations and settings.**

| Parameter | Value |
|---|---|
| Operating System/Simulator | Contiki OS/Cooja |
| Radio Medium Model | Unit Disk Graph Medium (UDGM): Distance Loss |
| Operating Carrier Frequency | 2.4 GHz |
| Channel Data Rate | 250 kbps |
| PAN Coordinator (Sink Node) | 1 |
| Network Layer | IPv6/6LoWPAN/RIME Stack |
| Routing Protocols | PSPCM, RPL |
| Number of Nodes for HPD and LPD Comparison | 300 |
| Number of Nodes for Benchmarking | 200 |
| Topology | DODAG Tree |
| Node Orientation | Random placement with no mobility |
| Propagation Mode | Single-hop and Multi-hop |
| Tx/Rx Range | 300m x 300m |
| Traffic Rate Type | Constant Bit Rate (CBR) |
| Simulation Time | 2000sec |
| IEEE 802.15.4 MTU | 127 bytes |
| Payload | 40 bytes |
| RF Transceiver | CC2420 |
| Transmission/Interference Range | 50m/100m |
| Microcontroller (MCU) | MSP430F2617 (Low-power) |
| PHY and MAC Protocol | IEEE 802.15.4 with CSMA/CA |
| Mote Type | Z1 (Zolertia) |

## 4.2. Performance metrics

The benchmarking of the PSPCM mechanism was performed on the following metrics;

- **Packet delivery ratio (PDR)**
  It is the ratio of the total packets received to total packets sent between source and destination nodes [3, 25]. PDR can be estimated by Eq 25.

$$PDR = \frac{TotalPacket_{ReceivedByDestination}}{TotalPackets_{SentBySourceNode}} \times 100\% \tag{25}$$

- **Throughput**
  It is the ratio of total packets received to the time taken to receive packets. It is measured in bytes per second or bits per second (bps) [42]. Throughput can be estimated by Eq 26.

$$Throughput = \frac{TotalPacket_{ReceivedByDestinationNode}}{TotalTime_{ToSendPacket}} \tag{26}$$

- **Power consumption**
  The operational longevity of the sensor network depends on how nodes utilize their limited energy/power as the network lifetime depends on energy/power consumption [6]. Energy and power are measured in millijoules (mJ)/ joules (J), and milliwatt (mW)/watt(W)

respectively. Node power consumption can be estimated by Eq 27.

$$Power_{Total} = Power_{Tx} + Power_{Rx} + Power_{mcu-idle/listen} + Power_{mcu-sleep} \tag{27}$$

- **Convergence time (CT)**
  This is a measure of how fast a group of nodes attains the state of convergence. Convergence is attained when all routing and topological information has been distributed to all nodes in the network. This time indicates how quickly a network constructs routing paths using ICMP control messages (DIS, DIO, DAO, and DAO-ACK) [26]. The quicker convergence is attained, the better the network latency and overall performance. CT is measured in time second (s) or millisecond (ms) and can be estimated by Eq 28. LCM represents the time for the Last Control Message and FCM represents the time for First Control Message.

$$CT = LCM_{(DIS/DIO/DAO/DAO-ACK)} - FCM_{(DIS/DIO/DAO/DAO-ACK)} \tag{28}$$

## 5. Results and discussions

The experimental results and analysis are presented in this section.

### 5.1. HPD and LPD performance comparison

To appreciate the data prioritization implementation improvement by PSPCM, it is imperative to compare the network performance between HPD and LPD. These performances were measured by the metrics presented in Subsection 4.2.

   **5.1.1. Packet Delivery Ratio (PDR)—HPD vs LPD.**   Fig 12 shows the PDR performance comparison between PSPCM—HPD, and PSPCM—LPD. The result shows a decreasing

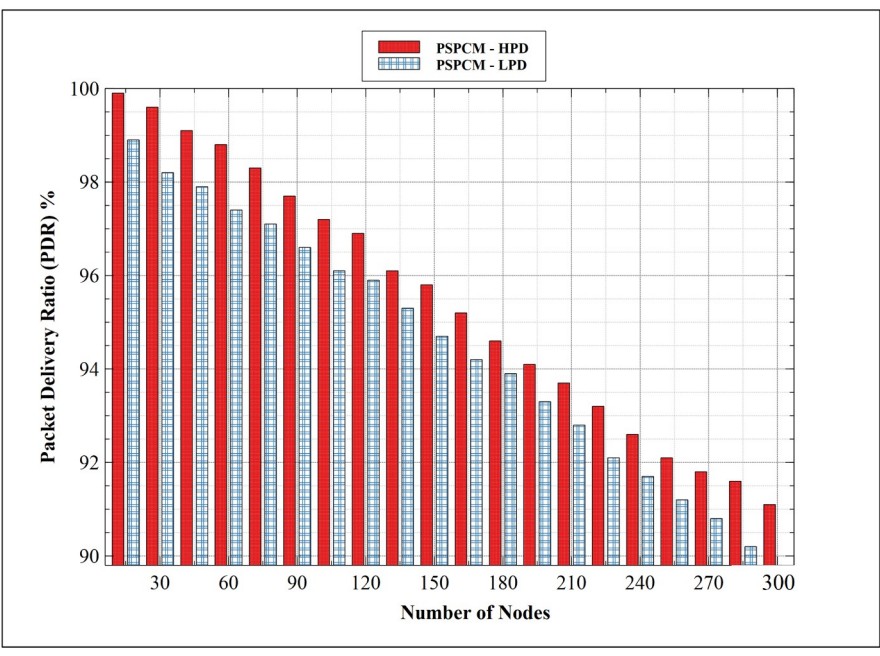

**Fig 12. Packet delivery ratio—HPD vs LPD.**

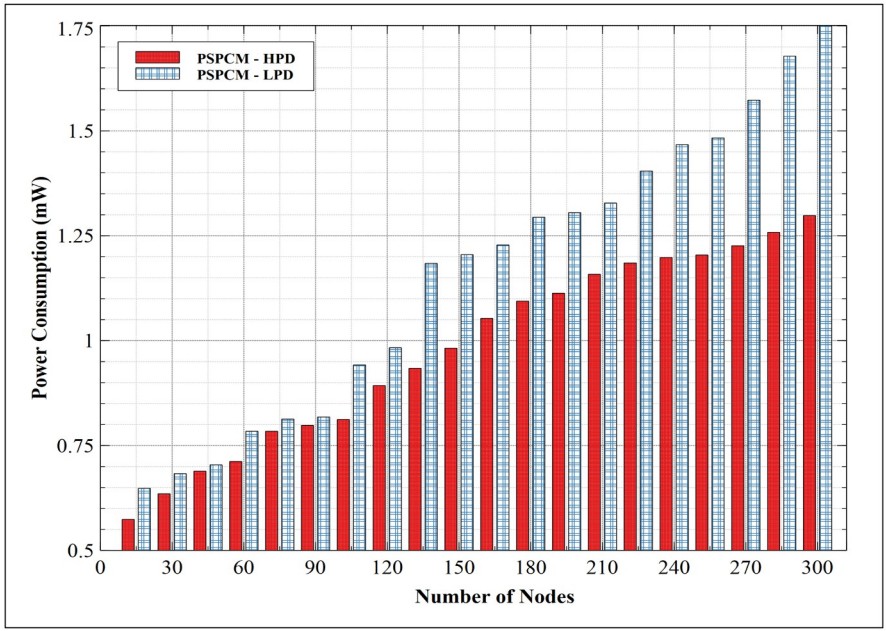

**Fig 13. Power consumption—HPD vs LPD.**

performance with an increasing number of nodes. PSPCM—HPD outperformed PSPCM—LPD. At 15 nodes, PSPCM—LPD presents its best performance of 98.9%, while PSPCM—HPD presents 99.9%. The performance of both priorities of data decreased uniformly until their least values of 89.8% and 91.1% for PSPCM—LPD, and PSPCM—HPD respectively at 300 nodes. The reduction in PDR performance is attributed to the increase in collision domain as the number of nodes and data packets involved in the network transaction increases, resulting in an increase in dropped packets. However, PSPCM—HPD presents a better PDR performance. The closer to 100% PDR a mechanism attains indicates a good performing network. The good performance of PSPCM—HPD is attributed to how the mechanism uses a more precise approach to select the appropriate and optimal parent, node ranks, and path resulting in the shortest possible path leading to the destination node from the source node.

**5.1.2. Power consumption—HPD vs LPD.** Fig 13 presents the power consumption of PSPCM—HPD, and PSPCM—LPD. Both data priorities show increased consumption with increasing nodes, which again is a result of an increased collision domain with an increasing number of data packets simultaneously involved in data transactions. PSPCM—HPD outperformed PSPCM—LPD. PSPCM—LPD presents its minimum power of 0.648mW at 15 nodes and 1.751mW at 300 nodes, while PSPCM—HPD presents 0.574mW minimum at 15 nodes and 1.298mW at 300 nodes. The better performance of PSPCM—HPD results from its precise way of estimating routing path to the sink node, of which once established, very minimum power is required to route data through the selected route to the sink.

**5.1.3. Throughput—HPD vs LPD.** Fig 14 shows the throughput performance between PSPCM—HPD, and PSPCM—LPD. Results show a decreasing throughput measure with increasing node number for both data classes, resulting from increased collision domain and consequently dropped packets. PSPCM—LPD exhibits its best performance of 9.17Kbps at 15 nodes and the least performance of 4.07Kbps at 300 nodes. At 15 nodes, PSPCM—HPD presents its best performance at a value of 10.35Kbps and the least value of 5.11Kbps at 300 nodes.

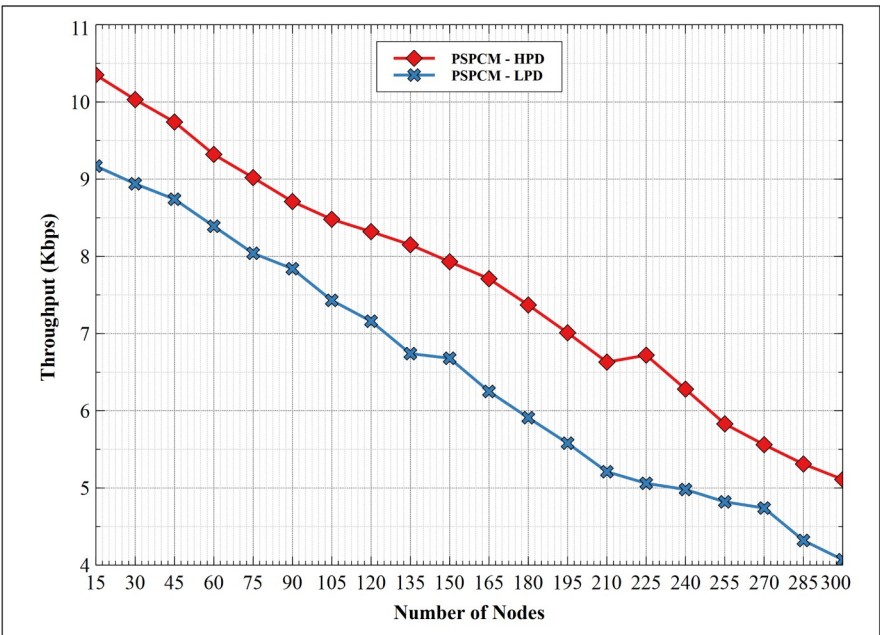

**Fig 14. Throughput—HPD vs LPD.**

The improvements of PSPCM—HPD is a result of how child-parent selection and routing path to the sink is implemented, which is more precise due to its time sensitivity.

**5.1.4. Convergence time—HPD vs LPD.** The creation of routing paths before the actual transportation of data packets across the network is a critical process, this is referred to as network convergence. After network initialization, the network must converge before data routing takes place and the time it takes to converge is crucial as it impacts the overall performance of the network. The lower the convergence time the better the network performance. Fig 15 presents the network convergence time between PSPCM—HPD and PSPCM—LPD. PSPCM—HPD and PSPCM—LPD show increasing convergence time as the number of nodes increases. At 15 nodes, PSPCM—LPD converges at 4.04 seconds at 15 nodes and 6.92 seconds at 300 nodes. On the other hand, PSPCM HPD converges at 3.23 seconds at 15 nodes and 5.93 seconds at 300 nodes. It is clear that PSPCM—HPD has better network convergence than PSPCM—LPD, this translates to lower network latency with improved network performance mostly in the favor of PSPCM—HPD.

## 5.2. Performance comparison with related mechanisms

The performance of the PSPCM mechanism with regard to priority-based routing mechanisms in the literature is considered in this section. These performances were measured by the performance metrics presented in Section 4.2.

**5.2.1. Packet Delivery Ratio (PDR).** Fig 16 shows the PDR between related mechanisms and PSPCM. All mechanisms show a decreasing performance with an increasing number of nodes, except for QCOF, which increased until the 90 nodes and decreased gradually till the 200 nodes. The decrease in PDR results from the increasing collision domain leading to an increasing number of packets dropped with the increasing number of nodes. At 10 nodes, QCOF presents the values of 12.70% and 13.90% at 200 nodes with its best performance of

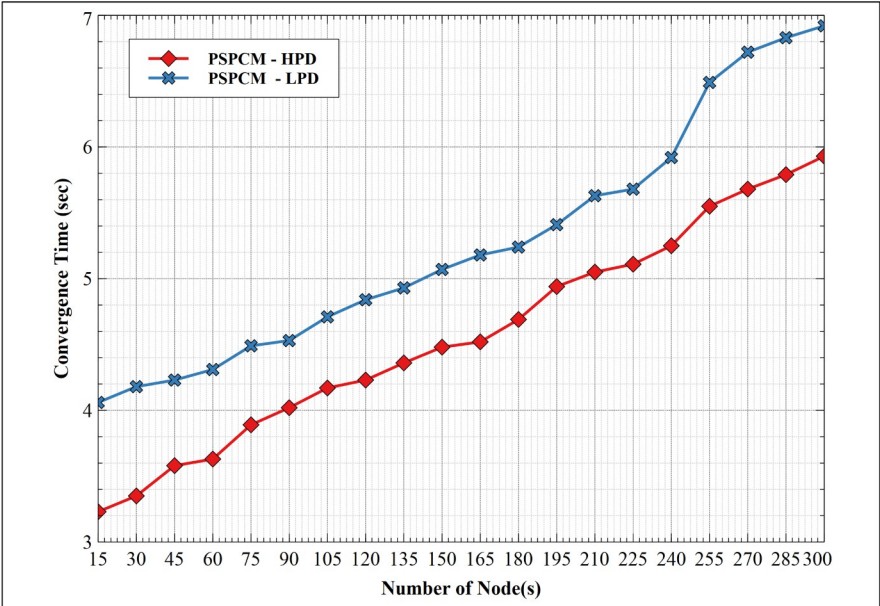

**Fig 15. Convergence time—HPD vs LPD.**

29.47% at 90 nodes. ETX presents its peak measure of 99.96% at 10 nodes, this value gradually decreased to a minimum of 16.32% at 200 nodes. This results from the complex mechanism used by ETX to establish a route to the sink which involves child-parent node selection. RPL-EL attained a maximum of 99.71% at the 10 nodes and decreased gradually to its

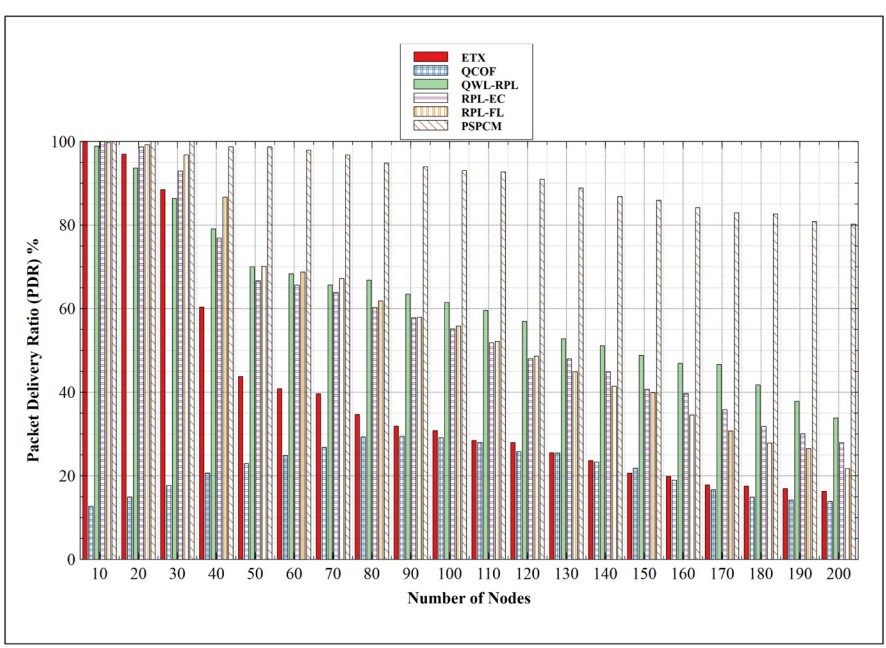

**Fig 16. Packet Delivery Ratio (PDR).**

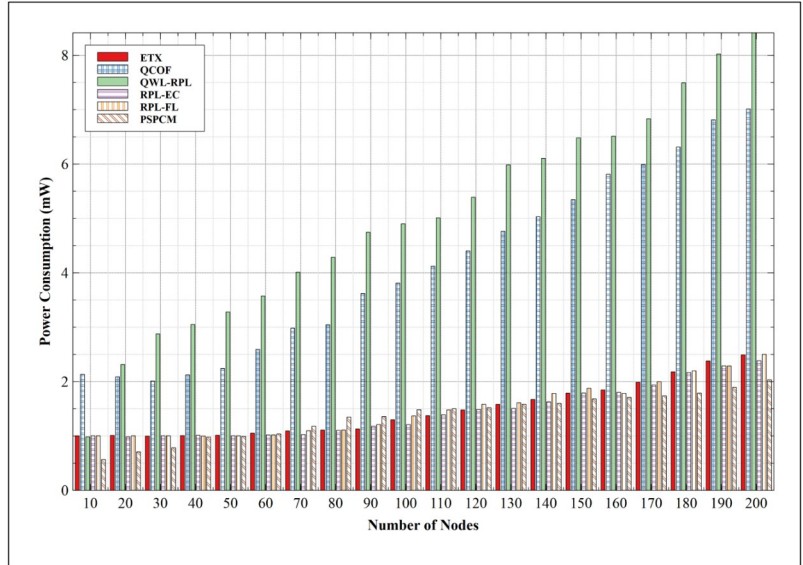

**Fig 17. Power consumption.**

minimum of 21.73% at the 200 nodes. This is closely followed by RPL-EC which has its maximum PDR of 99.95% at 15 nodes, with its minimum value of 27.92% at the 200 nodes. QWL-RPL registered decreasing PDR as node number increases, with its highest value at 98.89% at 10 nodes and 33.84% at the 200 nodes. Of all mechanisms, the proposed PSPCM mechanism has the best PDR performance values. PSPCM PDR value remained fairly high with a maximum value of 99.99% at the 10 nodes and a minimum of 80.22% at 200 nodes. Recalling that a network with PDR closer to 100% indicates a good network, hence the high PDR values of PSPCM shows that it has better support for scaling networks compared to other benchmarked mechanisms.

**5.2.2. Power consumption.** The efficient utilization of power is key to attaining a prolonged operational lifetime. Fig 17 presents the result of power consumption for all mechanisms. The result indicates a fairly consistent and gradual increase in power consumption for ETX, RPL-EC, and RPL-FL, and a sharp increase for QWL-RPL and QCOF. At maximum power consumption, ETX presents a value of 2.49mW at 200 nodes and the least value of 1.00mW at 10 nodes. This behavior is similarly exhibited by RPL-EC and RPL-FL. RPL-EC presents minimum consumption of 1.00mW at 10 nodes and maximum consumption of 2.38mW at 200 nodes. Subsequently, RPL-FL consumes 1.00mW at 10 nodes and peak values of 2.50mW at 200 nodes. These values guarantee gradual discharge of power, which supports a prolonged operational lifetime. QCOF consumes 2.13mW at 10 nodes and 7.01mW at 200 nodes, subsequently, QWL-RPL consumes the highest power of all mechanisms with the least consumption of 0.98mW at 10 nodes and peak consumption of 8.41mW at 200 nodes. This implies that nodes with QWL-RPL will be the first to run out of power compared to that of other mechanisms. Of all, PSPCM presents the most optimal power consumption mechanism, with its least values of 0.57mW at 10 nodes and 2.03mW at 200 nodes. This shows that PSPCM is more power-efficient and will exhibit an improved extended operational lifetime compare to the related mechanisms.

**5.2.3. Throughput.** Fig 18 presents the throughput of PSPCM in comparison to related mechanisms—ETX, QWL-RPL, QCOF, RPL-EC, and RPL-FL. Results show a steady increase

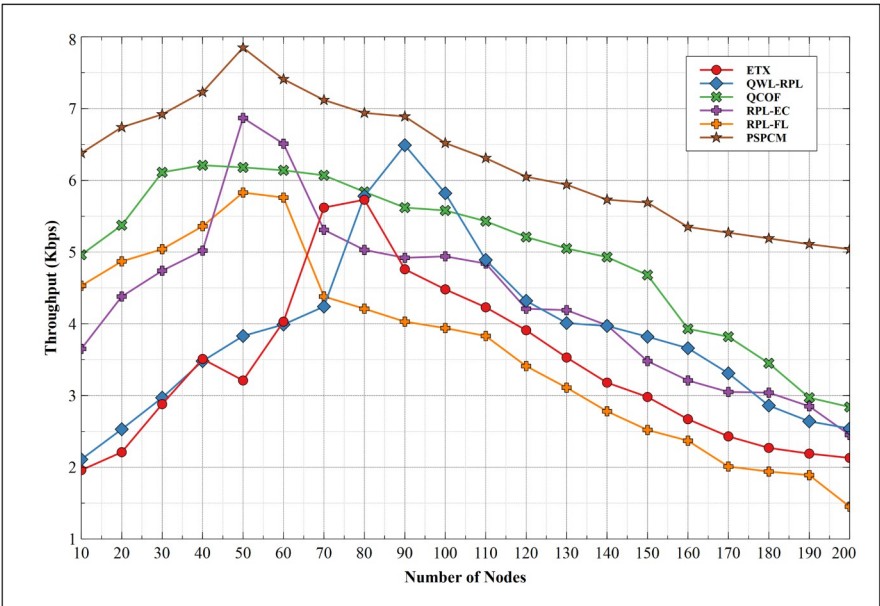

**Fig 18. Throughput.**

in the throughput with an increasing number of nodes until a point where the performance begins to drop as the number of nodes increases. This behavior is shown to be common among all mechanisms. The sudden decrease in throughput is a result of the increasing number of collision domains as the number of nodes and packets involved in data transactions increases. RPL-FL, started with a fairly good performance of 4.53Kbps at 10 nodes and rose to its best performance of 5.83Kbps at 50 nodes, is shown to have the least performance of 1.45Kbps at 200 nodes. This is followed by ETX, with its least values of 1.96Kbps at 10 nodes and peak value of 5.73Kbps at 80 nodes and finally drops to 2.13Kbps at 200 nodes. At 10 nodes, RPL-EC has its performance of 3.65Kbps and best performance of 6.87Kbps at 50 nodes, which finally dropped to 2.54Kbps at 200 nodes. This is followed by QWL-RPL with the values of 4.96Kbps at 10 nodes, 5.62Kbps at 90 nodes which are its best performance, and 2.84Kbps at 200 nodes. QCOF presents 2.11Kbps at 10 nodes, with a peak performance of 3.48Kbps at 40 nodes and the least performance of 2.54Kbps at 200 nodes. The proposed mechanism PSPCM presents the best performance compared to related mechanisms which start at 6.38Kbps at 10 nodes and the best performance of 7.85Kbps at 50 nodes and the least value of 5.04Kbps at 200 nodes. This improved performance results from the optimized shortest possible routes used by data packets to navigate the way to the destination node from the source nodes. The optimal route is also decided by the priority and class of the data.

**5.2.4. Convergence time.** Fig 19 shows the convergence time of related mechanisms in comparison to PSPCM. All mechanisms exhibit increasing convergence time as the number of nodes increases, this is a result of the increasing number of control messages as the number of nodes increases before network convergence is attained. QWL-RPL presents the highest convergence time with its minimum value of 9.34 seconds at 10 nodes and peak values of 81.74 seconds at 200 nodes. This is followed by ETX which shows a significantly high value of convergence time. ETX convergence occurs at 7.57 seconds at 10 nodes and a high of 37.82 seconds at the 200 nodes. Next is RPL-FL. At 10 nodes, RPL-FL converges at 4.92 seconds and

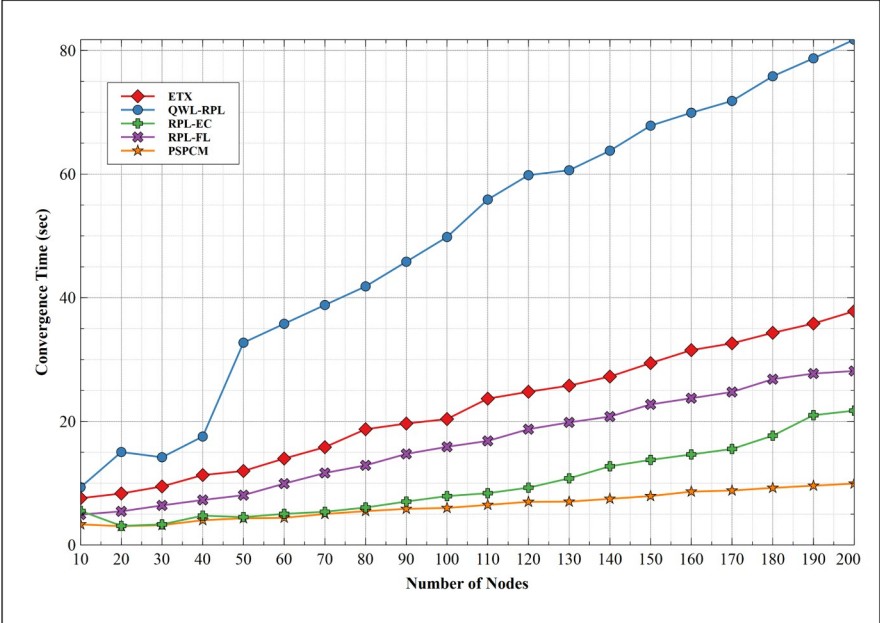

**Fig 19. Convergence time.**

28.15 seconds for 200 nodes. This is slightly improved compared to ETX. At the 10 nodes, RPL-EC has convergence at 5.52 seconds and 21.73 seconds at the 200 nodes. This indicates a better network performance compared to the first two mechanisms. QWL-RPL exhibits its highest convergence values with 9.34 seconds at 10 nodes and 32.73 seconds at 200 nodes. PSPCM presents improved network convergence with the least value of 3.34 seconds at 10 nodes and 9.93 seconds at 200 nodes. From the results, PSPCM demonstrates the best convergence time performance compared to the related mechanisms. This encourages improved latency and prolonged battery and network lifetime.

## 6. Conclusion and future work

The PSPCM mechanism presents an energy-efficient routing solution for heterogeneous data generated by WSN routed across the network layer with regards to the priorities convey by the data packet. The traditional parent selection, node ranking, and routing path mechanism of the standard RPL mechanism were improved to accommodate data priorities using the techniques of *class of service*. The *Reserve* field of the standard DIO base object of the RPL control message was modified by including the class of service tags. Subsequently, the standard RPL parent selections and routing path estimation mechanism were improved by the proposed *prioritized shortest path computation mechanism (PSPCM)*, which solves a single source shortest path problem as presented by the RPL DODAG routing graph, as well as problems presented by data heterogeneity. The advantage of the proposed mechanism is in its ability to take into consideration the heterogeneous data nature of WSN, classifying them and generating efficient and optimal shortest routing paths leading to the destination node while ensuring that the integrity of the network is not compromised. Validating the credibility of PSPCM, the mechanism was benchmarked against the standard RPL objective function—ETX and related mechanisms; QCOF, QWL-RPL, RPL-EC, and RPL-FL. Results showed that PSPCM outperformed all related mechanisms with an average PDR performance of 91.49%. Subsequently, the

PSPCM mechanism presented an average power consumption of 1.37mW across 200 nodes which is the best performance compared to all related mechanisms. This translates to a prolonged operational lifetime. Our future work will look into the node battery-level status for optimal parent selection for a distributed load-balanced WSN with support for an extended network lifetime.

## Supporting information

**S1 File.**
(ZIP)

## Acknowledgments

The authors would like to thank the editor and the anonymous reviewers for their meticulous approach to ensuring that this paper meets the required standard. Subsequently, this research would not have been thoroughly completed without the support provided by Universiti Teknologi Malaysia (UTM). The lab and academic resources have been very helpful. Thank you so much.

## Author Contributions

**Conceptualization:** Innocent Uzougbo Onwuegbuzie.

**Data curation:** Innocent Uzougbo Onwuegbuzie.

**Formal analysis:** Innocent Uzougbo Onwuegbuzie.

**Investigation:** Innocent Uzougbo Onwuegbuzie.

**Methodology:** Innocent Uzougbo Onwuegbuzie, Arafat Al-dhaqm.

**Resources:** Innocent Uzougbo Onwuegbuzie.

**Software:** Innocent Uzougbo Onwuegbuzie, Nor Badrul Anuar.

**Supervision:** Shukor Abd Razak.

**Validation:** Innocent Uzougbo Onwuegbuzie, Ismail Fauzi Isnin, Arafat Al-dhaqm.

**Visualization:** Innocent Uzougbo Onwuegbuzie.

**Writing – original draft:** Innocent Uzougbo Onwuegbuzie, Nor Badrul Anuar.

**Writing – review & editing:** Innocent Uzougbo Onwuegbuzie, Nor Badrul Anuar.

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
