## [Decision Letter · Decision Letter 0]

17 Feb 2021

PONE-D-20-34672

Prioritized Shortest Path Computation Mechanism (PSPCM) for Wireless Sensor Networks

PLOS ONE

Dear Dr. Al-dhaqm,

Thank you for submitting your manuscript to PLOS ONE. After careful consideration, we feel that it has merit but does not fully meet PLOS ONE’s publication criteria as it currently stands. Therefore, we invite you to submit a revised version of the manuscript that addresses the points raised during the review process.

We look forward to receiving your revised manuscript.

Kind regards,

Muhammad Asif, PhD

Academic Editor

PLOS ONE

Journal Requirements:

"This research was supported and funded by the University Teknologi Malaysia (UTM) with grant no. Q.J130000.2513.20H60."

"he author(s) received no specific funding for this work."

Additional Editor Comments:

Dear Authors

Thank you for your submission in Plos ONE, your manuscript has been review by the experts in the field and myself . we feel that it needs extensive revisions before we reconsider it. Therefore, Please revise the article in light of the reviewers comments and resubmit. Thank you

Reviewers' comments:

Reviewer's Responses to Questions

**Comments to the Author**

1. Is the manuscript technically sound, and do the data support the conclusions?

Reviewer #1: Yes

Reviewer #2: Yes

2. Has the statistical analysis been performed appropriately and rigorously? 

Reviewer #1: Yes

Reviewer #2: N/A

3. Have the authors made all data underlying the findings in their manuscript fully available?

Reviewer #1: Yes

Reviewer #2: Yes

4. Is the manuscript presented in an intelligible fashion and written in standard English?

Reviewer #1: Yes

Reviewer #2: Yes

5. Review Comments to the Author

Reviewer #1: This paper is about energy efficient routing of data in Wireless Sensor Networks. A problem well studied is moving sensed data to sink nodes. As the nodes have limited battery power, energy conserving routing of data is important. In the context of IoT a standard shortest path routing has been defined. The authors claim that priority of data should be considered in finding paths and develop a priority data based algorithm. They use a smart home as an application to demonstrate their approach. Essentially, they consider data to have either high priority or low priority. Routing is accomplished by setting up weighted graph and applying Dijkstra's shortest weighted path algorithm.

The paper describes the problem, discusses background and related work, present their approach, and then show simulation results compared to schemes that doesn't use priorities for data. As can be expected, by prioritizing they can improve performance of one class over another. They show results on delivery completion for high priority versus low priority data. They also compare the performance of PSPCM with other related algorithms and show improvements. Overall, the results seem to be interesting and useful. However, a more impactful work would be to understand the type of data and to see if some form of reduction in actual data bits sent, while maintaining full functionality, would conserve more energy.

The paper is well organized, however, careful reading is needed to ensure all English type errors are corrected.

Reviewer #2: This manuscript proposes a prioritized shortest path computation mechanism (PSPCM) by introducing high priority data (HPD) and Low priority data (LPD) metric (value of 0 or 1) within the data package (DIO message object). Their results show improved performance (Packet delivery rate and power consumption) compared to other techniques (ETX, QCOF, QWL-RPL, RPL-EC, RPL-FL).

Comments:

1) Results for Throughput (Sec. 5.1.3; Fig. 11) only shows performance within proposed PSPCM (i.e. HPD vs LPD). These must also be compared with other techniques (i.e. ETX, QCOF, QWL-RPL, RPL-EC, RPL-FL) without priority to show that the proposed technique is superior or similar in performance.

2) The proposed scheme uses binary values of HPD and LPD. It is not clear what criteria will be used by the source to decide on this binary decision. Greedy sources might set all their packages to HPD, as there is no objective measure or criteria, which will result in loss of advantages of the proposed scheme.

3) If the packet delivery ratio for LPD is very low for 100 nodes (Fig. 9), what will happen to the Fig. 13 if the number of nodes is increased to 100? The same argument does for Fig. 10, 11, 12 also. Thus, for Sec. 12, all simulation needs to be updated to at least 100 nodes. As most practical network is much larger to this, it begs the question of the number of nodes is very large, then will the proposed PSPCM will lose its advantage?

4) As LPD packages will be less prioritized, then there is an issue of inability to guarantee Quality of Service (QoS) for these packages. How this can be resolved?

5) 5G wireless network system, which is upcoming, resolves a lot of existing issues with high priority data and latency. The authors need to compare their results with 5G protocols.

6. PLOS authors have the option to publish the peer review history of their article (what does this mean?). If published, this will include your full peer review and any attached files.

Reviewer #1: No

Reviewer #2: No

---

## [Author Response · Author response to Decision Letter 0]

2 Apr 2021

Innocent Uzougbo ONWUEGBUZIE

School of Computing, Faculty of Engineering,

Universiti Teknologi Malaysia (UTM)

Johor Bahru, Johor, Malaysia

31th March 2021.

The Editor,

PLoS ONE

Dear Editor,

Rebuttal Letter

Thank you for inviting us to submit a revised draft of our manuscript entitled, "Prioritized shortest path computation mechanism (PSPCM) for wireless sensor networks" for consideration by PLoS ONE. We also appreciate the time and effort you and each of the reviewers have dedicated to providing insightful feedback on ways to strengthen our paper. Thus, it is with great pleasure that we resubmit our revised paper for further consideration. We have incorporated changes that reflect the detailed suggestions you have meticulously provided. We also hope that our updates/improvements and the responses we provide below satisfactorily address all the issues and concerns you and the reviewers have raised.

For clarity, we have presented our responses to your comments and questions in a Section format; Section 1: Editors' comment and response, Section 2: Response to the evaluation by Reviewer 1, and Section 3: Response to the evaluation by Reviewer 2.

Section 1: Editors' comment and response

Response: 

Thank you for your comments.

The paper has been formatted to meet PLOS ONE’s style requirements.

Section 2: Response to the evaluation by Reviewer 1

1. Overall, the results seem to be interesting and useful. However, a more impactful work would be to understand the type of data and to see if some form of reduction in actual data bits sent, while maintaining full functionality, would conserve more energy. 

Response:

Thank you so much for your insightful comments and suggestions to improve the quality of our work. 

The type of data traversed across the network is in bits and bytes (8 bits equals a byte). By standard, the IEEE 802.15.4 allows for a maximum transmission unit (MTU) of 127 bytes; 25 bytes for MAC Frame Header, and 102 bytes for payload. However, of the 102 bytes for payload, 21 bytes are used for link-layer security, 40 bytes for IPv6 header, and 8 bytes for UDP header. Eventually, the payload is left with only 33 bytes. Depending on the size of data sensed from the monitored environment, the 33 bytes may not be enough. To increase the payload size, the techniques of Header Compression (HC) were used to compress the IPv6 header and UDP header from 48 bytes to 6 bytes. This eventually increased the payload capacity to 75 bytes. For this research, a payload size of 40 bytes was used. This was sufficient enough to ensure the optimal performance of the proposed PSPCM with conserved energy utilization. Further details of this are presented in the third and fourth paragraphs of Section 3.1.

The entire simulation/experimentation was reperformed with the updated parameters. To also show support for PSPCM network scalability, the number of nodes for HPD and LPD comparison was increased from 100 to 300, while the number of nodes for comparison with related mechanisms (i.e. ETX, QCOF, QWL-RPL, RPL-EC, RPL-FL), was increased from 50 to 200. The new findings are discussed in Sections 5.1 and 5.2

2. The paper is well organized; however, careful reading is needed to ensure all English-type errors are corrected. 

Response:

Thank you for your comments.

The entire paper has been proofread, and all grammatical and spelling errors have been corrected.

Section 2: Response to the evaluation by Reviewer 2.

1. Results for Throughput (Sec. 5.1.3; Fig. 11) only shows performance within the proposed PSPCM (i.e. HPD vs LPD). These must also be compared with other techniques (i.e. ETX, QCOF, QWL-RPL, RPL-EC, RPL-FL) without priority to show that the proposed technique is superior or similar in performance. 

Response:

Thank you very much for your comments.

The Throughput performance of PSPCM in comparison with related mechanisms (i.e. ETX, QCOF, QWL-RPL, RPL-EC, RPL-FL) is now presented and supported with updated discussion in Section 5.2.3. The result shows that PSPCM outperformed compared mechanisms.

2. The proposed scheme uses binary values of HPD and LPD. It is not clear what criteria will be used by the source to decide on this binary decision. Greedy sources might set all their packages to HPD, as there is no objective measure or criteria, which will result in a loss of advantages of the proposed scheme. 

Response:

Thank you for your comments.

The deciding factor is the Class of Service (CS) tag of the data, which is pre-configured into the sensor nodes (High Priority Node – HPN and Low Priority Node – LPN) before network operations begin. See Fig. 4. The 8-bits Reserved field of the DIO base object which runs in the network layer of the WSN protocol stack of all nodes is modified to accommodate data of varying priorities and this is represented by the CS tags, where bits 00000000 is for CS = 0 representing Low Priority Data (LPD) and bits 00000001 is for CS = 1 representing High Priority Data (HPD). See Fig. 7. In order words, HPN is pre-configured with CS = 1, and LPN are pre-configured with CS = 0 before the commencement of network operations. In this way, it becomes impossible for any of the node types to behave greedily in favor of its class of data. As either of the data class traverses the network, intermediate or relay nodes computes the shortest traversal distance between source and destination node. Further details are presented in Sections 3.1 and 3.1.1.

3. If the packet delivery ratio for LPD is very low for 100 nodes (Fig. 9), what will happen to Fig. 13 if the number of nodes is increased to 100? The same argument does for Fig. 10, 11, 12 also. Thus, for Sec. 12, all simulation needs to be updated to at least 100 nodes. As the most practical network is much larger than this, it begs the question of the number of nodes is very large, then will the proposed PSPCM will lose its advantage? 

Response:

Thank you for your comments and question.

The number of nodes has been increased from 50 to 200 for comparison with related mechanisms (i.e. ETX, QCOF, QWL-RPL, RPL-EC, RPL-FL) and the simulations/experimentations was reperformed, results was used to plot the graphs - Figs 16, 17, 18 and 19 for PDR, Power consumption, Throughput, and Convergence time respectively. The findings is supported by discussion in Section 5.2. Subsequently, to show support for network scalability of the proposed mechanism – PSPCM, and its HPD and LPD comparison, the number of simulated nodes was increased from 100 to 300, and the simulations/experimentations were reperformed. The obtained results were used to plot the graphs – Figs 12, 13, 14, and 15, for PDR, Power consumption, Throughput, and Convergence time respectively. A detailed discussion of the findings and is presented in Section 5.1.

4. As LPD packages will be less prioritized, then there is an issue of inability to guarantee Quality of Service (QoS) for these packages. How this can be resolved? 

Response:

Thank you for your comments and question.

A much clearer explanation as to how PSPCM works have been presented in Section 3.1.6. The applications of the mechanism have also been presented in Figs. 3 and 4 (updated/improved) of Section 3.1. In this Figure, it is shown that High Priority Nodes (HPN), which generates HPD are installed in locations whose data are deemed critical, time-sensitive, and of high priority, likewise, Low Priority Nodes (LPN) which generates LPD are installed in locations whose data are less critical delay-tolerant and low priority. HPN and LPN are preconfigured and assigned a Class of Service (CS) of 1 and 0 respectively. This is implemented in the Reserved field of the modified DIO Base Object as presented in Fig 7 (updated) with explanations provided in Sections 3.1.1, while Section 3.1.6 presents the details of how the proposed PSPCM works. With this approach, data emanating from either HPN or LPN are routed to the sink using Eqs 14 and 16 for HPD or Eqs 15 and 17 for LPD. This approach helps to check and prevent any data from acting greedily to the detriment of the other data class, thereby preserving and improving the Quality of Service (QoS).

5. 5G wireless network system, which is upcoming, resolves a lot of existing issues with high priority data and latency. The authors need to compare their results with 5G protocols. 

Response:

Thank you very much for your comments.

While 5G is gradually gaining the center-stage particularly for high-speed internet, with very minimal latency, at the moment, it will not completely wipe-out the existing technologies like 4G and WSN. One major drawback of 5G is the high cost of implementation and high power/energy consumption, which currently does not seem good for low-power and lossy network (LLN) devices. On the other hand, LLN devices have the advantage of very low power consumption making it possible for them to be installed in just any location and left unattended for several months or years without battery replacement. To make the best of both technologies, 5G and WSN can be implemented in conjunction together in a multi-tiered network, where the lower-tier is for WSN (data generation and aggregation) and the upper-tier is for transporting the aggregated data by high-speed 5G network for either immediate processing and or storage. This is what we implemented in our approach. Further detail of this is presented in Fig. 3, and Section 3.1.

Again, thank you for giving us the opportunity to strengthen our manuscript with your valuable comments and queries. We have worked hard to incorporate your feedback and hope that these revisions persuade you to accept our submission.

Sincerely yours,

Innocent Uzougbo ONWUEGBUZIE

(Corresponding Author)

Universiti Teknologi Malaysia (UTM)

+601160770749

---

## [Decision Letter · Decision Letter 1]

9 Jun 2021

PONE-D-20-34672R1

Prioritized Shortest Path Computation Mechanism (PSPCM) for Wireless Sensor Networks

PLOS ONE

Dear Dr. Onwuegbuzie,

Thank you for submitting your manuscript to PLOS ONE. After careful consideration, we feel that it has merit but does not fully meet PLOS ONE’s publication criteria as it currently stands. Therefore, we invite you to submit a revised version of the manuscript that addresses the points raised during the review process.

We look forward to receiving your revised manuscript.

Kind regards,

Chakchai So-In, Ph.D.

Academic Editor

PLOS ONE

Reviewers' comments:

Reviewer's Responses to Questions

**Comments to the Author**

1. If the authors have adequately addressed your comments raised in a previous round of review and you feel that this manuscript is now acceptable for publication, you may indicate that here to bypass the “Comments to the Author” section, enter your conflict of interest statement in the “Confidential to Editor” section, and submit your "Accept" recommendation.

Reviewer #2: All comments have been addressed

Reviewer #3: All comments have been addressed

2. Is the manuscript technically sound, and do the data support the conclusions?

Reviewer #2: Yes

Reviewer #3: Partly

3. Has the statistical analysis been performed appropriately and rigorously? 

Reviewer #2: N/A

Reviewer #3: Yes

4. Have the authors made all data underlying the findings in their manuscript fully available?

Reviewer #2: Yes

Reviewer #3: Yes

5. Is the manuscript presented in an intelligible fashion and written in standard English?

Reviewer #2: Yes

Reviewer #3: No

6. Review Comments to the Author

Reviewer #2: The revised manuscript has addressed my past concerns. I have no further comments for this manuscript.

Reviewer #3: The problems which were proposed in the last version were carefully checked and revised, however, some fetal problems exist.

1. The organization of the manuscript should be improved to highlight the objective, method, and main contribution.

2. The author is expected to demonstrate the main advantage of the current method.

3. Part 2 is to be improved, and it could be more concise.

7. PLOS authors have the option to publish the peer review history of their article (what does this mean?). If published, this will include your full peer review and any attached files.

Reviewer #2: No

Reviewer #3: No

---

## [Author Response · Author response to Decision Letter 1]

2 Aug 2021

Reviewers’ Comments Responses

1. The manuscript should be written in standard English, all grammar and typographical errors should be corrected Thank you so much for your comments.

All grammatical and typographical errors in the article has been corrected.

2. The organization of the manuscript should be improved to highlight the objective, method, and main contribution Thank you for your comments.

The organization of the article has been improved to highlight the objective, method and main contribution. This is presented in the 6th paragraph of Section 1: Introduction.

3. The author is expected to demonstrate the main advantage of the current method. Thank you for your comments.

The main advantage of the proposed mechanism is in its ability to take into consideration the heterogenous data nature of WSN, classifying them and generating efficient and optimal shortest routing paths leading to the destination node (sink) while ensuring that the integrity of the network is not compromised. This detail is presented in paragraph six (6) of Section 1: Introduction, and Section 6 of Conclusion and future work

4. Part 2 is to be improved, and it could be more concise. Thank you for your comments.

Part 2 of the article has been improved with the review of recent literatures in a concise manner.

---

## [Decision Letter · Decision Letter 2]

18 Oct 2021

PONE-D-20-34672R2Prioritized Shortest Path Computation Mechanism (PSPCM) for Wireless Sensor NetworksPLOS ONE

Dear Dr. Onwuegbuzie,

Thank you for submitting your manuscript to PLOS ONE. After careful consideration, we feel that it has merit but does not fully meet PLOS ONE’s publication criteria as it currently stands. Therefore, we invite you to submit a revised version of the manuscript that addresses the points raised during the review process.

Kindly take a look at the comment with minor decision as follows: Please submit your revised manuscript by Dec 02 2021 11:59PM. If you will need more time than this to complete your revisions, please reply to this message or contact the journal office at plosone@plos.org. Please include the following items when submitting your revised manuscript:A rebuttal letter that responds to each point raised by the academic editor and reviewer(s). You should upload this letter as a separate file labeled 'Response to Reviewers'.A marked-up copy of your manuscript that highlights changes made to the original version. You should upload this as a separate file labeled 'Revised Manuscript with Track Changes'.An unmarked version of your revised paper without tracked changes. You should upload this as a separate file labeled 'Manuscript'.If applicable, we recommend that you deposit your laboratory protocols in protocols.io to enhance the reproducibility of your results. Protocols.io assigns your protocol its own identifier (DOI) so that it can be cited independently in the future. For instructions see: https://journals.plos.org/plosone/s/submission-guidelines#loc-laboratory-protocols. Additionally, PLOS ONE offers an option for publishing peer-reviewed Lab Protocol articles, which describe protocols hosted on protocols.io. Read more information on sharing protocols at https://plos.org/protocols?utm_medium=editorial-email&utm_source=authorletters&utm_campaign=protocols.

We look forward to receiving your revised manuscript.

Kind regards,

Chakchai So-In, Ph.D.

Academic Editor

PLOS ONE

Journal Requirements:

Reviewers' comments:

Reviewer's Responses to Questions

**Comments to the Author**

1. If the authors have adequately addressed your comments raised in a previous round of review and you feel that this manuscript is now acceptable for publication, you may indicate that here to bypass the “Comments to the Author” section, enter your conflict of interest statement in the “Confidential to Editor” section, and submit your "Accept" recommendation.

Reviewer #3: All comments have been addressed

Reviewer #4: (No Response)

2. Is the manuscript technically sound, and do the data support the conclusions?

Reviewer #3: Yes

Reviewer #4: Yes

3. Has the statistical analysis been performed appropriately and rigorously? 

Reviewer #3: Yes

Reviewer #4: (No Response)

4. Have the authors made all data underlying the findings in their manuscript fully available?

Reviewer #3: Yes

Reviewer #4: (No Response)

5. Is the manuscript presented in an intelligible fashion and written in standard English?

Reviewer #3: Yes

Reviewer #4: (No Response)

6. Review Comments to the Author

Reviewer #3: The problems I have proposed in the last version were carefully checked and revised, the quality of the paper seems fine. I recommend its publication.

Reviewer #4: This paper studied the "Prioritized shortest path computation mechanism (PSPCM) for wireless sensor networks ". The quality should be improved. Minor revision should be done for this version of the paper as follows:

*More achievements on this topic should be added for the Section "introductions" and "Related work" .

*Mathematics modeling to analyze the method is not enough.

* Some references are out-of-date, so these references before 2014 should be deleted. At the same time, many important recent references are missing, which can support the idea of this paper, the following references should be added in the Section "References":

1- "Multipath routing through the firefly algorithm and fuzzy logic in wireless sensor networks". Peer-to-Peer Networking and Applications, 14(2), 541-558.

2- "An energy-aware clustering and two-level routing method in wireless sensor networks". Computing, 102(7), 1653-1671.

3- "A hierarchical secure data aggregation method using the dragonfly algorithm in wireless sensor networks". Peer-to-Peer Networking and Applications, 1-26.

4- "A reliable tree-based data aggregation method in wireless sensor networks". Peer-to-Peer Networking and Applications, 14(2), 873-887.

5- "SHSDA: secure hybrid structure data aggregation method in wireless sensor networks". Journal of Ambient Intelligence and Humanized Computing, 1-20.

6- "EELRP: energy efficient layered routing protocol in wireless sensor networks". Computing, 1-21.

7- "A method for routing and data aggregating in cluster‐based wireless sensor networks". International Journal of Communication Systems, 34(7), e4754.

*The writing format of the paper should be revised.

* The format of some references in this paper does not completed.

7. PLOS authors have the option to publish the peer review history of their article (what does this mean?). If published, this will include your full peer review and any attached files.

Reviewer #3: **Yes: **Haochun Zhang

Reviewer #4: No

---

## [Author Response · Author response to Decision Letter 2]

10 Dec 2021

Responses to comments by Reviewer 4

1. More achievements on this topic should be added for the Section "introductions" and "Related work" 

Thank you for your comments and suggestions.

More achievements have been added to the “Introduction” and “Related work” sections, and their references have been included in the References section.

2. Mathematics modeling to analyze the method is not enough. 

Thank you for your comments.

The mathematics modeling to analyze the proposed mechanism has been improved and presented in a more detailed manner. This is shown in Section 3.1.5

3. Some references are out-of-date, so these references before 2014 should be deleted. At the same time, many important recent references are missing, which can support the idea of this paper, the following references should be added in the Section "References" 

Thank you for your comments and suggestions.

All out-of-date references before 2014 have been removed and updated with up-to-date references as suggested. Please refer to the References section.

4. The writing format of the paper should be revised. 

Thank you for your comments and suggestions.

The writing format of this paper has been revised as suggested.

5. The format of some references in this paper does not completed. 

Thank you for your comments.

References with incomplete format have been completely removed and updated with properly formatted and up-to-date references. Kindly refer to the References section.

---

## [Decision Letter · Decision Letter 3]

16 Feb 2022

Prioritized Shortest Path Computation Mechanism (PSPCM) for Wireless Sensor Networks

PONE-D-20-34672R3

Dear Dr. Onwuegbuzie,

We’re pleased to inform you that your manuscript has been judged scientifically suitable for publication and will be formally accepted for publication once it meets all outstanding technical requirements.

Kind regards,

Chakchai So-In, Ph.D.

Academic Editor

PLOS ONE

Additional Editor Comments (optional):

Thank you for submitting the paper for PLOS ONE and further follow the the process toward publication process. 

C.S.

Reviewers' comments:

Reviewer's Responses to Questions

**Comments to the Author**

1. If the authors have adequately addressed your comments raised in a previous round of review and you feel that this manuscript is now acceptable for publication, you may indicate that here to bypass the “Comments to the Author” section, enter your conflict of interest statement in the “Confidential to Editor” section, and submit your "Accept" recommendation.

Reviewer #4: All comments have been addressed

2. Is the manuscript technically sound, and do the data support the conclusions?

Reviewer #4: Yes

3. Has the statistical analysis been performed appropriately and rigorously? 

Reviewer #4: Yes

4. Have the authors made all data underlying the findings in their manuscript fully available?

Reviewer #4: No

5. Is the manuscript presented in an intelligible fashion and written in standard English?

Reviewer #4: Yes

6. Review Comments to the Author

Reviewer #4: Necessary corrections have been made in the article and the authors have answered reviewer's comments. Now, the quality of the paper is suitable.

7. PLOS authors have the option to publish the peer review history of their article (what does this mean?). If published, this will include your full peer review and any attached files.

Reviewer #4: No

---

## [Editor Report · Acceptance letter]

22 Feb 2022

PONE-D-20-34672R3 

Prioritized shortest path computation mechanism (PSPCM) for wireless sensor networks 

Dear Dr. Onwuegbuzie:

I'm pleased to inform you that your manuscript has been deemed suitable for publication in PLOS ONE. Congratulations! Your manuscript is now with our production department. 

Kind regards, 

on behalf of

Dr. Chakchai So-In 

Academic Editor

PLOS ONE